# Airway *Prevotella* promote TLR2-dependent neutrophil activation and rapid clearance of *Streptococcus pneumoniae* from the lung

Kadi J. Horn[1], Melissa A. Schopper[1], Zoe G. Drigot[1,2] & Sarah E. Clark [1✉]

This study investigates how specific members of the lung microbiome influence the early immune response to infection. *Prevotella* species are a major component of the endogenous airway microbiota. Increased abundance of *Prevotella melaninogenica* correlates with reduced infection with the bacterial pathogen *Streptococcus pneumoniae*, indicating a potentially beneficial role. Here, we show that *P. melaninogenica* enhances protection against *S. pneumoniae*, resulting in rapid pathogen clearance from the lung and improved survival in a mouse lung co-infection model. This response requires recognition of *P. melaninogenica* lipoproteins by toll-like receptor (TLR)2, the induction of TNFα, and neutrophils, as the loss of any of these factors abrogates *Prevotella*-induced protection. Improved clearance of *S. pneumoniae* is associated with increased serine protease-mediated killing by lung neutrophils and restraint of *P. melaninogenica*-induced inflammation by IL-10 in co-infected mice. Together, these findings highlight innate immune priming by airway *Prevotella* as an important protective feature in the respiratory tract.

---

[1] University of Colorado School of Medicine, Department of Otolaryngology, Aurora, CO 80045, USA. [2] University of Colorado Boulder, College of Arts and Sciences, Boulder, CO 80309, USA. ✉email: sarah.e.clark@cuanschutz.edu

The respiratory tract is home to a diverse microbial community that varies in composition by anatomical location and over the course of life. *Prevotella* are frequently identified as one of the top three most abundant bacteria detected in the oral cavity and lungs of healthy adults[1–4]. While the largest biomass and most stable microbial communities in the respiratory tract are located in the upper airway, the lung is constantly exposed to these bacteria by micro-aspiration. Bacteria from the oral cavity are the dominant source of lung microbiome "seeding"[1,2]. In the absence of the steep oxygen gradients that arise during chronic lung infection, sustained growth of obligate anaerobes such as *Prevotella* in the lung remains debated, but frequent exposures may nevertheless function as an important immune regulatory signal.

The purpose of this study was to determine how airway *Prevotella* influence the early immune response to infection with the bacterial pathogen *Streptococcus pneumoniae* (the pneumococcus). *S. pneumoniae* is the most common cause of community-acquired pneumonia, which has an estimated economic impact of $17 billion in the U.S. alone and is the leading cause of death in children under 5 years old[5,6]. Pneumococcal vaccines are extremely effective against invasive disease, but are not as protective against pneumonia, and the phenomenon of serotype replacement has increased the number of non-vaccine strains in circulation[7]. Asymptomatic colonization of the nasopharynx with *S. pneumoniae* is a prerequisite for disease development, following invasion to other tissues including the lung. However, it remains unclear why some individuals are more predisposed to lung infection with *S. pneumoniae* compared with others. The microbiome is thought to help protect against bacterial lung infections, which are worsened in antibiotic treated and germ-free mice[8,9]. *S. pneumoniae* also persists more readily following antibiotic therapy[10]. *Prevotella* are a candidate "protective" member of the airway microbiome in the context of *S. pneumoniae* infection, as several studies indicate an association between increased abundance of *Prevotella* and reduced *S. pneumoniae* in both the upper airway[11,12] and lung[2,13,14]. Similarly, *Prevotella* species including *P. melaninogenica* are more abundant in healthy adults than those with pneumonia[15,16]. While species-level classification in such analyses is limited, in one report *P. melaninogenica* was ranked as the most discriminative operational taxonomic unit (OTU) differentiating healthy adults from patients with pneumococcal pneumonia[12]. *P. melaninogenica* is an important member of the endogenous airway microbiota, as it is part of the core oral microbiome[17,18] and selectively enriched in the lung[1,4,19]. The factors underlying the potential benefit of airway *Prevotella* on protection against pneumococcal pneumonia are unresolved.

In healthy adults, the enrichment of *Prevotella* rRNA gene concentration in bronchoalveolar lavage fluid (BAL) is associated with sub-clinical inflammation, including increased numbers of activated neutrophils[20]. However, *Prevotella*-induced inflammation is subdued compared to that induced by opportunistic bacterial pathogens. For example, lung transplant recipients with *Prevotella*-dominant BAL microbiome profiles had lower innate immune activation profiles than those with *Staphylococcus* or *Pseudomonas*-dominant microbiomes[21]. In mice, lung exposure to *Prevotella melaninogenica* induced less toll-like receptor(TLR) 2-dependent inflammation than the pathogen *Haemophilus influenzae*[22]. Similarly, another *Prevotella* species induced TLR2 signaling but fewer pro-inflammatory cytokines than a cystic fibrosis (CF) pathogen in human CF bronchial epithelial cells[23]. In the context of CF, both positive and negative roles have been attributed to anaerobic bacteria including *P. melaninogenica*, though *Prevotella* abundance correlates with milder CF in some cases[24]. *Prevotella* species are not universally beneficial or benign

members of the airway microbiome. Unlike *P. melaninogenica*, *Prevotella intermedia* is frequently isolated from periodontal abscesses, expresses virulence factors including a type 6 secretion system (T6SS), and enhances susceptibility to *S. pneumoniae* by upregulating a critical host adherence receptor[25,26]. *P. melaninogenica* has been identified in polymicrobial lung abscesses, though it is unclear which species drive abscess formation, and causes increased tissue damage in the context of bleomycin-induced pulmonary fibrosis in mice[27]. These studies suggest that the overall beneficial versus detrimental effects of *Prevotella* on airway health depend on the amplitude and context of *Prevotella*-induced immune responses.

For this study, we developed an animal model to determine how airway *Prevotella* modulate innate immune-mediated protection against *S. pneumoniae*. Using this model, we identify an innate immune response induced by several *Prevotella* species including *P. melaninogenica* associated with rapid clearance of *S. pneumoniae* from the lung, highlighting airway *Prevotella* as an important immune regulatory signature that contributes to respiratory tract health.

## Results

**P. melaninogenica enhances clearance of S. pneumoniae.** While the murine airway microbiome contains both *Prevotella* and *Streptococcus* species, the strains used in these studies are not resident members, allowing us to control exposures to each bacterium. *Prevotella* aspiration was modeled in mice by instilling *P. melaninogenica* intratracheally (i.t.) prior to challenge with *S. pneumoniae* (Fig. 1a). *S. pneumoniae* burdens at 24 h post-infection were not detected in any mice pre-exposed to an equivalent dose of live *P. melaninogenica*, compared to burdens of ~$10^5$ colony forming units (CFUs) in mice infected with *S. pneumoniae* alone (Fig. 1b). Titration of *P. melaninogenica* revealed that significant protection was maintained with up to a 100-fold lower dose of live *P. melaninogenica* than that of *S. pneumoniae* (Fig. 1b). Burdens of $10^5$ CFU/mL *P. melaninogenica* have been reported in human lung lavage fluid[19], which falls within this protective range. These data suggest that exposure to live *P. melaninogenica* mediates rapid clearance of *S. pneumoniae* from the murine lung.

We next examined the impact of inactivated (heat-killed, HK) *P. melaninogenica* on survival following a lethal dose of *S. pneumoniae*. Exposure to *P. melaninogenica* significantly increased the probability of survival, with 33% of *Prevotella*-exposed mice succumbing to infection compared to 85% of mice infected with *S. pneumoniae* alone (Fig. 1c). Improved survival in mice exposed to *P. melaninogenica* correlated with lower pneumococcal burdens in the lung 3 days post-infection (Supplementary Fig. 1a). The rapid protection induced by *P. melaninogenica* was not dependent on instillation of *Prevotella* into the lung, as intranasal inoculation similarly enhanced *S. pneumoniae* clearance by 24 h (Supplementary Fig. 1b). *P. melaninogenica* was protective against both serotype 2 *S. pneumoniae*, which spreads systemically by 24 h, and serotype 3 *S. pneumoniae*, which is restricted to the lung (Fig. 1c, d). Together, these data indicate that exposure to live or inactivated *P. melaninogenica* protects against *S. pneumoniae* infection.

We next determined whether instillation of any HK bacterium enhances protection against *S. pneumoniae*. Initially, we investigated other HK bacteria which are common members of the upper airway microbiota and are known to mediate direct competition with *S. pneumoniae*, including two *Corynebacterium* species[28,29] and *Streptococcus salivarius*[30]. In contrast to live strains, two of which reduce *S. pneumoniae* infection in a similar pre-exposure model[29], none of the HK strains were protective

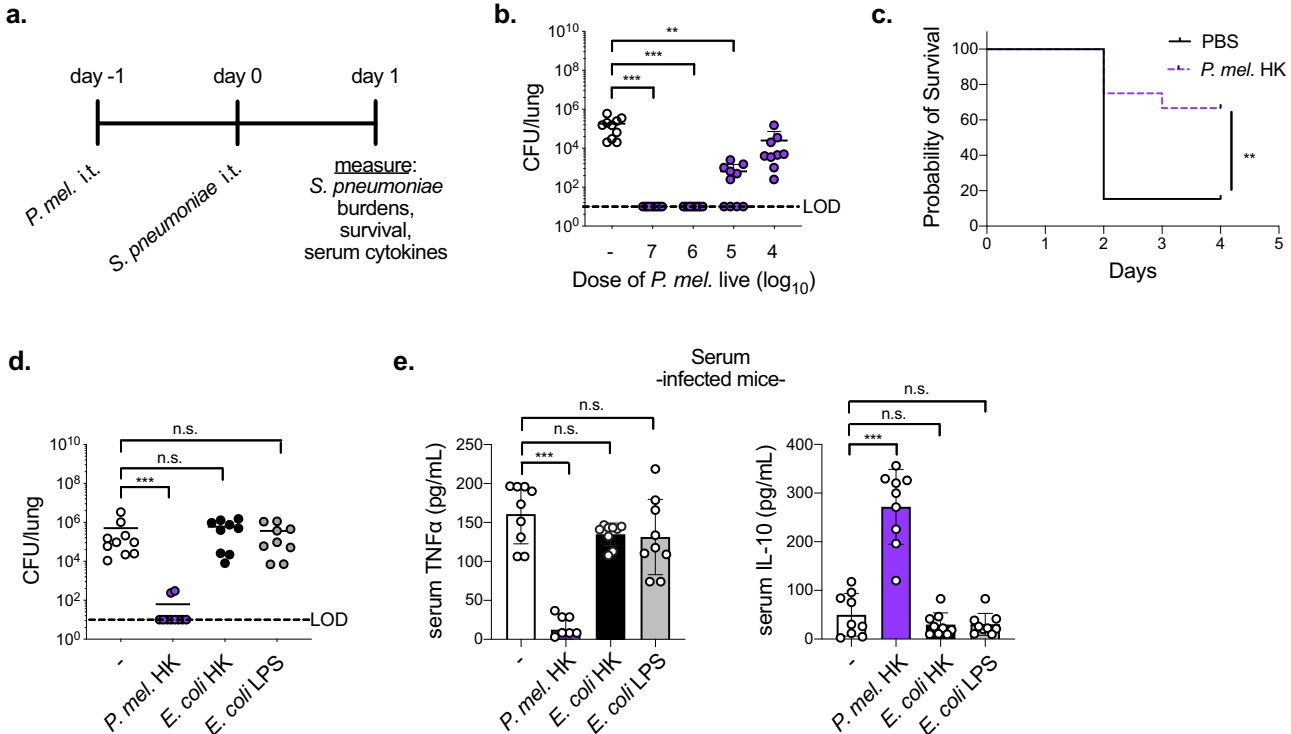

**Fig. 1 Exposure to airway *P. melaninogenica* (*P. mel.*) induces rapid clearance of *S. pneumoniae*. a** Timeline for *Prevotella* exposure followed by *S. pneumoniae* infection. **b** Lung type 2 *S. pneumoniae* burdens following exposure to PBS (-) or live *P. mel.* strain 25845 i.t. at indicated doses prior to 24 h infection with *S. pneumoniae*, $5×10^6$ CFU/mouse ($n = 10$ mice/group). **c** Survival following exposure to PBS or *P. mel.* heat-killed (HK, $10^7$ CFU equivalents/mouse) i.t. prior to *S. pneumoniae* infection, $10^7$ CFU/mouse ($n = 13$ mice/group). **d**, **e** Lung *S. pneumoniae* burdens (**d**) and serum cytokines (**e**) in mice following exposure to PBS (-), *P. mel.* HK, *E. coli* HK, or *E. coli* lipopolysaccharide (LPS) i.t. prior to 24 h *S. pneumoniae* infection, $5×10^6$ CFU/mouse ($n = 10$ mice/group). LOD = limit of detection. Data are pooled from three independent experiments, displayed as mean ± SEM. For **b** from left to right ***$p < 0.0001$, ***$p < 0.0001$, **$p = 0.0015$, Kruskal–Wallis with Dunn's post hoc test, **c** **$p = 0.0075$, Mantel-Cox survival test, **d** from left to right ***$p = 0.0008$, $p > 0.9999$, $p > 0.9999$, Kruskal–Wallis with Dunn's post hoc test, **e** from left to right ***$p < 0.0001$, $p = 0.2352$, $p = 0.1558$ (TNFα), from left to right ***$p < 0.0001$, $p = 0.6977$, $p = 0.7136$ (IL-10), one-way ANOVA with Dunnett's post hoc test. Source data are provided as a Source Data file.

(Supplementary Fig. 1e), indicating a unique effect mediated by *P. melaninogenica* HK compared with these airway commensals, which like *S. pneumoniae* are Gram-positive. *P. melaninogenica* is a Gram-negative bacterium, and the lipopolysaccharide (LPS) of Gram-negative bacteria is an abundant TLR4 agonist. We next compared the impact of exposure to Gram-negative *Escherichia coli* HK as well as *E. coli* LPS, which improves protection against other lung pathogens by immune priming[31,32]. Unlike *P. melaninogenica*, neither *E. coli* HK nor *E. coli* LPS enhanced clearance of *S. pneumoniae* from the lung (Fig. 1d), suggesting that not all Gram-negative bacteria have a similar effect. Systemic cytokine responses also differed in mice exposed to *P. melaninogenica* compared with *E. coli* HK or *E. coli* LPS (Fig. 1e). These findings indicate that *P. melaninogenica*, but not *E. coli* LPS, enhances protection against *S. pneumoniae* lung infection, allowing for the use of *E. coli* LPS as an immune stimulatory but non-protective comparative tool.

**P. melaninogenica activates lung neutrophils.** The rapid protective effect of *P. melaninogenica* implies a role for the innate immune system. We focused on the use of *P. melaninogenica* HK to investigate the mechanism of this effect, as burdens of live *P. melaninogenica* may themselves be altered by the disruption of innate immune pathways. The immune response induced by *P. melaninogenica* HK was first examined in the absence of *S. pneumoniae* infection. *P. melaninogenica* exposure increased the production of several pro-inflammatory cytokines in lung bronchoalveolar lavage fluid (BAL), including TNFα, IL-6, IL-1α,

and IFNγ as well as the chemokines MCP-1 (CCL2) and MIP-2 (CXCL2), a major neutrophil chemoattractant[33] (Fig. 2a). *P. melaninogenica* also increased systemic TNFα and IL-10 compared to mice treated with PBS or *E. coli* LPS (Fig. 2b). The recruitment and activation of innate immune cells in the lung was evaluated by intracellular flow cytometry (Supplementary Fig. 2a). *P. melaninogenica* significantly enhanced the recruitment of myeloid cells including inflammatory monocytes and neutrophils, similar to *E. coli* LPS (Fig. 2c, d and Supplementary Fig. 2b, c). However, unlike *E. coli* LPS, *P. melaninogenica* also induced TNFα production in lung neutrophils (Fig. 2e). Neither stimulus affected the recruitment or TNFα production of CD11b^hi dendritic cells (DCs) (Supplementary Fig. 2d, e). These data indicate that exposure to *P. melaninogenica* induces a pro-inflammatory response in the lung associated with increased neutrophil recruitment and activation.

*P. melaninogenica*-exposed mice had fewer alveolar macrophages (AMs) in their lungs than mice exposed to either PBS or *E. coli* LPS (Supplementary Fig. 2f). Others have reported that *E. coli* LPS activates AM production of TNFα, which we confirm, though this response was not induced by *P. melaninogenica* (Supplementary Fig. 2g). These findings indicate distinct lung innate immune cell activation profiles for mice exposed to *E. coli* LPS compared with *P. melaninogenica*, which unlike *E. coli* LPS is protective against *S. pneumoniae*.

**Neutrophils and TNFα are critical for *P. melaninogenica* protection.** Neutrophils are important for *S. pneumoniae* killing at

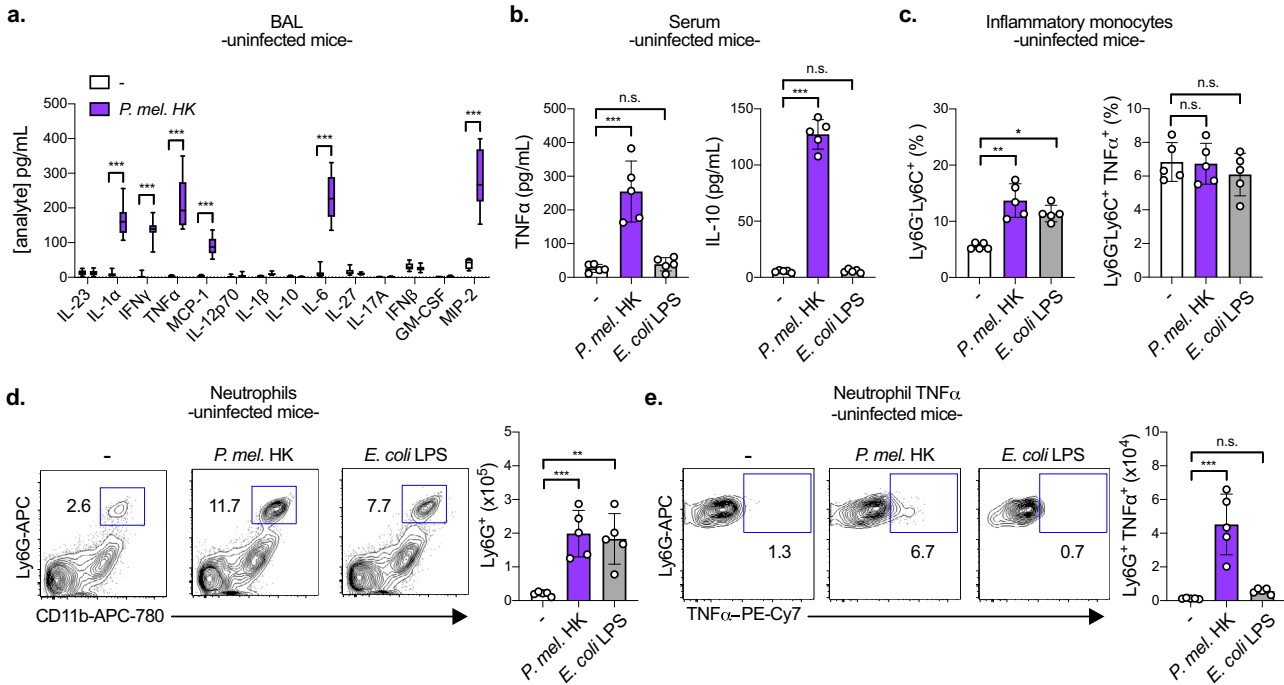

**Fig. 2 *P. melaninogenica* (*P. mel.*) induces a pro-inflammatory immune response in the lung associated with neutrophil activation in the absence of *S. pneumoniae* infection. a** Quantification of cytokines and chemokines in lung bronchoalveolar lavage (BAL) following exposure to PBS (-) or *P. mel.* heat-killed (HK) strain 25845 i.t. for 24 h ($n = 10$ mice/group). **b** Serum cytokines detected in mice exposed to PBS (-), *P. mel.* HK, or *E. coli* lipopolysaccharide (LPS) i.t. for 24 h ($n = 5$ mice/group). **c–e** Percentage and total number of inflammatory monocytes (CD45$^+$SiglecF$^-$Ly6G$^-$Ly6C$^+$CD11b$^+$ cells) (**c**), neutrophils (CD45$^+$SiglecF$^-$Ly6G$^+$CD11b$^+$ cells) (**d**), and neutrophil TNFα (**e**) detected in the lungs of mice from **b** by intracellular flow cytometry, with representative plots shown for the percentage of neutrophils and neutrophil TNFα ($n = 5$ mice/group). Data are pooled from three independent experiments (**a**) or representative from one of four independent experiments (**b–e**). Box boundaries in **a** indicate the 25th and 75th percentiles, with a horizontal line representing the median and whiskers indicating minimum and maximum values. Data in **b–e** are displayed as mean ± SEM. For **a** ***$p < 0.0001$, two-way ANOVA with Sidak's post hoc test, **b** from left to right ***$p < 0.0001$, $p = 0.9108$ (TNFα), ***$p < 0.0001$, $p = 0.9108$ (IL-10), one-way ANOVA with Dunnett's post hoc test and from left to right $p = 0.0060$, $p > 0.9999$ (TNFα), $p = 0.0144$, $p > 0.9999$ (IL-10), Kruskal–Wallis with Dunn's post hoc test, **c** from left to right **$p = 0.0044$, *$p = 0.0453$, $p = 0.9875$, $p = 0.5395$, Kruskal–Wallis with Dunn's post hoc test, **d** from left to right ***$p = 0.009$, **$p = 0.0017$, one-way ANOVA with Dunnett's post hoc test and $p = 0.0115$, $p = 0.0215$, Kruskal–Wallis with Dunn's post hoc test, **e** from left to right ***$p < 0.0001$, $p = 0.7296$, one-way ANOVA with Dunnett's post hoc test and $p = 0.0008$, $p = 0.1542$, Kruskal–Wallis with Dunn's post hoc test. Source data are provided as a Source Data file.

early time points, prior to the development of specific immunity, which is typically required for clearance[34]. Exposure to *P. melaninogenica* HK increased the number of neutrophils recruited to the lungs in response to *S. pneumoniae* infection (Fig. 3a). To address the contribution of neutrophils to pneumococcal clearance in *Prevotella*-exposed mice, we depleted neutrophils prior to *S. pneumoniae* infection. In the absence of neutrophils, *P. melaninogenica* was no longer protective against *S. pneumoniae*, as lung burdens in *Prevotella*-exposed mice were similar to those in mice infected with *S. pneumoniae* alone (Fig. 3b). These data indicate that neutrophils are required for *P. melaninogenica*-mediated protection against *S. pneumoniae*.

*P. melaninogenica*-induced TNFα was significantly reduced in the lungs of neutrophil-depleted mice, indicating neutrophils as a major source of TNFα in this setting (Supplementary Fig. 2h). We next considered whether the pro-inflammatory response associated with elevated neutrophil TNFα is required for *P. melaninogenica*-mediated protection by depleting TNFα. Neutrophil recruitment and production of TNFα were reduced in TNFα-depleted mice (Fig. 3c, d, and Supplementary Fig. 3a, b). Similar to the effect of neutrophil depletion, TNFα depletion resulted in loss of *P. melaninogenica*-mediated protection (Fig. 3e). Inflammatory monocyte recruitment was also reduced in TNFα depleted mice, though CD11b$^{hi}$ DCs and AM populations were unchanged (Supplementary Fig. 3c–e). Together, these data indicate a requirement for both

neutrophils and TNFα in *P. melaninogenica*-mediated protection against *S. pneumoniae*.

**P. melaninogenica is protective in microbiome-depleted mice.** While our findings suggest that innate immune priming is important for *P. melaninogenica*-mediated protection, it was unclear whether the endogenous microbiome, which regulates immune homeostasis, is required for this protective effect. The contribution of the endogenous microbiome to protection in *Prevotella*-exposed mice was evaluated using antibiotic treated and Germ-free mice. In antibiotic treated mice, live *P. melaninogenica* significantly improved *S. pneumoniae* clearance from the lungs by 24 h (Fig. 4a and Supplementary Fig. 4a). Similarly, exposure to live *P. melaninogenica* enhanced clearance of *S. pneumoniae* from the lungs of Germ-free mice (Fig. 4b). In Germ-free mice, *Prevotella*-mediated protection against *S. pneumoniae* was associated with increased lung neutrophil recruitment and activation (Fig. 4c and Supplementary Fig. 4b). These data indicate that *P. melaninogenica* is sufficient to improve protection against *S. pneumoniae* lung infection in the absence of an intact microbiome.

**P. melaninogenica lipoproteins induce TLR2-dependent neutrophil TNFα.** We next investigated the host and bacterial requirements for *P. melaninogenica*-induced neutrophil TNFα

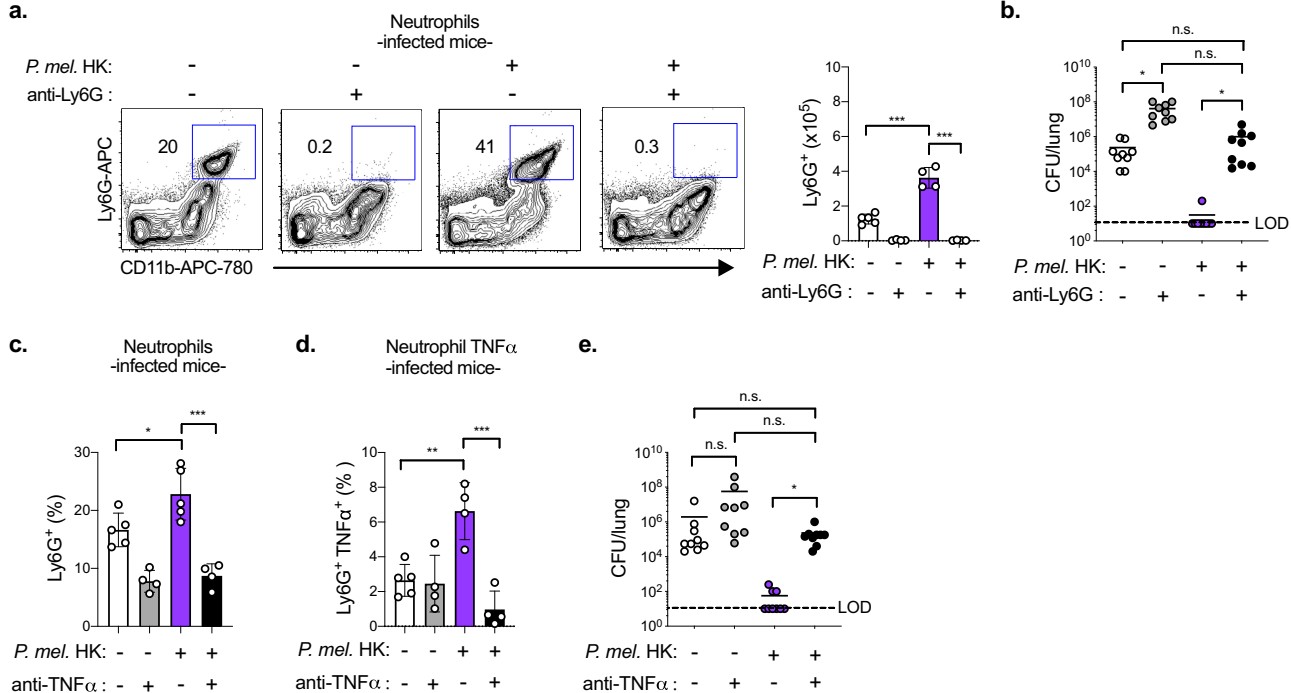

**Fig. 3 Neutrophils and TNFα are critical for *P. melaninogenica* (*P. mel.*)-mediated protection against *S. pneumoniae* infection. a, b** Representative flow cytometry plots and total cell numbers for neutrophils detected by flow cytometry (*n* = 5 mice/group, PBS, n = 4 mice/group, others) (**a**) and lung burdens of type 2 *S. pneumoniae* (*n* = 9 mice/group) (**b**) in mice treated with isotype control or anti-Ly6G antibodies (200 μg/mouse) i.p. together with either PBS (-) or *P. mel.* strain 25845 heat-killed (HK) i.t. prior to 24 h *S. pneumoniae* infection, 5×10⁶ CFU/mouse. **c–e** Total cell number of neutrophils and neutrophil TNFα detected by intracellular flow cytometry (*n* = 5 mice/group for isotype control groups, 4 mice/group for anti-TNFα groups) (**c, d**) and lung *S. pneumoniae* burdens (*n* = 9 mice/group) (**e**) in mice treated with isotype control or anti-TNFα antibodies (200 μg/mouse) i.p. together with either PBS (-) or *P. mel.* HK i.t. prior to 24 h *S. pneumoniae* infection. LOD = limit of detection. Data are pooled from three independent experiments (**b, e**) or representative from one of four independent experiments (**a, c, d**), displayed as mean ± SEM. For **a** ***$p < 0.0001$, one-way ANOVA with Sidak's post hoc test, **b** from left to right *$p = 0.0222$, $p > 0.9999$, $p = 0.0751$, *$p = 0.0192$, Kruskal–Wallis with Dunn's post hoc test, **c** from left to right *$p = 0.0337$, ***$p < 0.0001$ one-way ANOVA with Tukey's post hoc test, **d** **$p = 0.003$, ***$p = 0.0002$ one-way ANOVA with Tukey's post hoc test, **e** from left to right $p = 0.3058$, $p > 0.9999$, $p = 0.4470$, *$p = 0.0118$, Kruskal–Wallis with Dunn's post hoc test. Source data are provided as a Source Data file.

in vitro. In neutrophils purified from the bone marrow of naïve mice, exposure to *P. melaninogenica* HK induced TNFα secretion in a dose-dependent manner (Fig. 5a). Small molecule inhibitors of TLR2 (C29)[35] and TLR4 (TAK-242)[36] were used to compare the importance of TLR2 versus TLR4 for *P. melaninogenica*-induced TNFα. Inhibition of TLR2, but not TLR4, resulted in loss of *P. melaninogenica*-induced neutrophil TNFα (Fig. 5b), suggesting that TLR2 signaling is important for this response. In support of this conclusion, *P. melaninogenica*-induced TNFα secretion was absent in neutrophils purified from TLR2 deficient (*Tlr2⁻/⁻*) mice, in contrast to neutrophils from wild-type (WT) mice (Fig. 5c). These data indicate that TLR2 is required for *P. melaninogenica* stimulation of neutrophil TNFα secretion.

Lipoproteins, a component of the cell membrane in both Gram-negative and Gram-positive bacteria, are the primary bacterial TLR2 ligand[37]. In contrast, TLR4 is responsive to LPS[38]. We purified lipoproteins and LPS from *P. melaninogenica* and found that *P. melaninogenica* lipoproteins, but not *P. melaninogenica* LPS, induced neutrophil TNFα secretion in a TLR2-dependent manner (Fig. 5c). The digestion of bacterial lipoproteins with lipoprotein lipase abrogates TLR2 activation[39]. We found that lipoprotein lipase digestion of *P. melaninogenica* resulted in significant loss of neutrophil TNFα secretion in vitro (Fig. 5d). Lipoprotein lipase-treated *P. melaninogenica* was also no longer protective against *S. pneumoniae* infection in mice (Fig. 5e). However, lipoprotein-TLR2 signaling was not sufficient for protection, as neither the TLR2 agonist Pam3SK4 nor purified *P. melaninogenica* lipoproteins altered lung burdens of *S.*

*pneumoniae* compared to mice treated with PBS (Fig. 5f). Together, these findings indicate that *P. melaninogenica* lipoproteins activate TLR2-dependent secretion of TNFα in neutrophils and are required, but not sufficient, for *P. melaninogenica*-mediated protection against *S. pneumoniae*.

**TLR2 is required for *P. melaninogenica*-mediated protection.**
We next addressed the importance of TLR2 for *P. melaninogenica*-induced immune activation and protection in vivo. In *Tlr2⁻/⁻* mice exposed to *P. melaninogenica* HK alone (uninfected), the recruitment of both neutrophils and inflammatory monocytes was similar to that in WT mice (Fig. 6a and Supplementary Fig. 5a, b), and CD11b^hi DC and AM populations were unaffected (Supplementary Fig. 5c, d). However, *P. melaninogenica*-induced neutrophil TNFα was lost in *Tlr2⁻/⁻* mice (Fig. 6b). These findings indicate that TLR2 is required for *P. melaninogenica*-induced neutrophil TNFα, but not *P. melaninogenica*-induced myeloid cell recruitment.

In *Tlr2⁻/⁻* mice infected with *S. pneumoniae*, lung burdens were elevated regardless of *Prevotella* exposure, in contrast to WT mice in which *Prevotella* exposure was protective (Fig. 6c). *P. melaninogenica*-induced neutrophil recruitment was lost in *Tlr2⁻/⁻* mice infected with *S. pneumoniae*, while the recruitment of inflammatory monocytes was maintained (Fig. 6d, e and Supplementary Fig. 5e, f). These data indicate that *Prevotella*-induced neutrophil recruitment is TLR2-dependent in *S. pneumoniae*-infected mice, a setting which may require TNFα

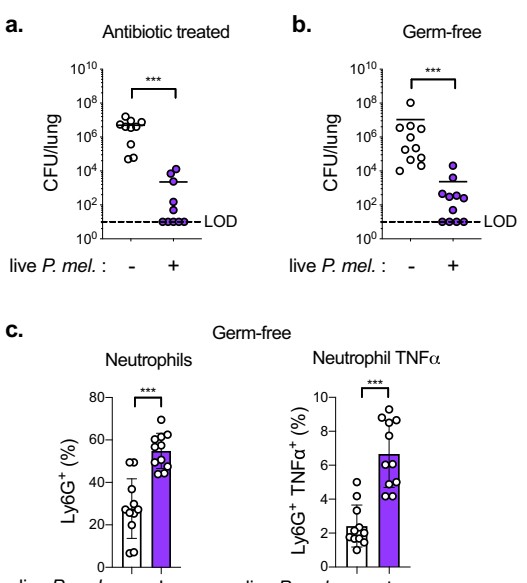

**Fig. 4 P. melaninogenica (P. mel.) is protective against S. pneumoniae infection in microbiome-depleted mice. a** Lung *S. pneumoniae* burdens in mice treated with antibiotics followed by exposure to PBS (-) or live *P. mel.* strain 25845 i.t. prior to 24 h *S. pneumoniae* infection, $5 \times 10^6$ CFU/mouse (*n* = 9 mice/group). **b, c** Lung *S. pneumoniae* burdens (**b**) and percent neutrophils, neutrophil TNFα (**c**) in Germ-free mice treated with either PBS (-) or live *P. mel.* i.t. prior to 24 h *S. pneumoniae* infection, $10^6$ CFU/mouse (*n* = 11 mice/group). LOD = limit of detection. Data are pooled from three independent experiments. Data are displayed as mean ± SEM. For **a**, **b** ***$p < 0.0001$ two-tailed Mann–Whitney *U*-test, **c** ***$p < 0.0001$ two-tailed *t*-test. Source data are provided as a Source Data file.

feedback. Neutrophils in the lungs of $Tlr2^{-/-}$ infected mice also no longer expressed TNFα in response to *P. melaninogenica* (Fig. 6f). The proportions of other lung myeloid cells, including CD11b$^{hi}$ DCs and AMs, were similar between WT and $Tlr2^{-/-}$ mice infected with *S. pneumoniae* (Supplementary Fig. 5g, h). Together, these data indicate a critical role for TLR2 in *P. melaninogenica*-induced neutrophil activation and protection against *S. pneumoniae*. The separation between neutrophil expression of TNFα, which was TLR2-dependent, and the recruitment of neutrophils and inflammatory monocytes, which were TLR2-independent in *P. melaninogenica*-exposed mice, indicate that myeloid cell recruitment is by itself insufficient to improve protection against *S. pneumoniae*, which requires the activation of cells including neutrophils.

**P. melaninogenica enhances serine protease-mediated killing.** To address whether neutrophils activated by *P. melaninogenica* directly contribute to enhanced clearance of *S. pneumoniae*, neutrophil killing of *S. pneumoniae* was measured in vitro. Neutrophils purified from the bone marrow of naïve mice were no better at killing *S. pneumoniae* following pre-incubation with *P. melaninogenica* HK for up to 6 h (Supplementary Fig. 6a), after which the viability of primary neutrophils declines. This suggests that while direct exposure to *P. melaninogenica* induces neutrophil secretion of TNFα, it is not sufficient to enhance killing of *S. pneumoniae* within this timeframe.

We next considered whether additional cells or signals in the lungs of *P. melaninogenica*-exposed mice promote neutrophil killing of *S. pneumoniae*. To address this, neutrophils were purified from the lungs of mice exposed to either *P. melaninogenica* or *E. coli* LPS for 24 h (Fig. 7a). *E. coli* LPS was chosen as a

non-protective immune stimulus that would still induce neutrophil recruitment. Neutrophils purified from the lungs of *Prevotella*-exposed mice killed significantly more *S. pneumoniae* than neutrophils purified from the lungs of mice exposed to *E. coli* LPS (Fig. 7b). The two primary mechanisms for neutrophil killing of *S. pneumoniae* are the production of reactive oxygen species (ROS) and activity of serine proteases. Neutrophils from *P. melaninogenica*-exposed and *E. coli* LPS-exposed mice produced similar levels of total ROS (Fig. 7c). In contrast, the activity of two serine proteases, cathepsin G and elastase, was significantly higher in neutrophils purified from the lungs of *P. melaninogenica*-exposed mice, compared to neutrophils from the lungs of mice exposed to *E. coli* LPS (Fig. 7d). Neutrophils pooled from naïve mice had similarly low serine protease activity as those from *E. coli* LPS-exposed mice (Supplementary Fig. 6b). The addition of protease inhibitors, but not the ROS inhibitor DPI, significantly reduced *S. pneumoniae* killing by neutrophils isolated from *P. melaninogenica*-exposed mice (Fig. 7e), without affecting *S. pneumoniae* growth (Supplementary Fig. 6c). These data indicate that serine protease activity is important for *P. melaninogenica*-enhanced killing in lung neutrophils.

The importance of TLR2 signaling for *Prevotella*-induced neutrophil killing was determined by comparing neutrophils purified from WT versus $Tlr2^{-/-}$ mice. Neutrophils from the lungs of $Tlr2^{-/-}$ mice exposed to *P. melaninogenica* were less efficient at killing *S. pneumoniae* compared to those from WT mice (Fig. 7f) and these cells had negligible serine protease activity (Fig. 7g), demonstrating a critical role for TLR2 in *P. melaninogenica*-enhanced neutrophil killing. Together, these findings suggest that *P. melaninogenica* exposure increases TLR2-dependent, serine protease-mediated killing of *S. pneumoniae* by lung neutrophils.

**P. melaninogenica protection requires regulation by IL-10.** Immune regulation is critical to mitigate the damaging effects of inflammation in the lung such as barrier disruption and reduced oxygen exchange. While TNFα primes several protective immune responses, overproduction causes tissue damage and impairs *S. pneumoniae* clearance[40,41]. The anti-inflammatory cytokine IL-10 is a master regulator of pro-inflammatory responses including TNFα[42]. In purified neutrophils, we found that in addition to TNFα, *P. melaninogenica* HK induced the secretion of IL-10 in a dose-dependent and TLR2-dependent manner (Supplementary Fig. 7a). Further, neutrophil secretion of TNFα was inhibited in cultures exposed to both *P. melaninogenica* and *S. pneumoniae*, suggesting regulation of this response (Supplementary Fig. 7b). These findings mirror the systemic TNFα response in *Prevotella*-exposed mice, which was reduced following *S. pneumoniae* infection. Additionally, *P. melaninogenica*-exposed mice infected with *S. pneumoniae* had significantly reduced levels of several pro-inflammatory cytokines in lung BAL, including TNFα, IL-6, IL-1α, IFNβ, and IFNγ, compared to those infected with *S. pneumoniae* but not exposed to *Prevotella* (Fig. 8a). This is in contrast to uninfected mice, where *P. melaninogenica* exposure increased BAL pro-inflammatory cytokines at 24 h, indicating that by 48 h in co-infected mice, these responses were reduced. These data suggest that exposure to *P. melaninogenica* regulates *S. pneumoniae*-induced inflammation in the lung.

To address the potential role of regulatory IL-10 in *P. melaninogenica*-exposed mice, we compared *S. pneumoniae* burdens in WT and IL-10 deficient ($Il10^{-/-}$) mice with and without pre-exposure to *P. melaninogenica* HK. Unexpectedly, *P. melaninogenica*-mediated protection against *S. pneumoniae* was lost in $Il10^{-/-}$ mice, which had similarly high burdens regardless of *P. melaninogenica* exposure (Fig. 8b). In $Il10^{-/-}$ mice, serum

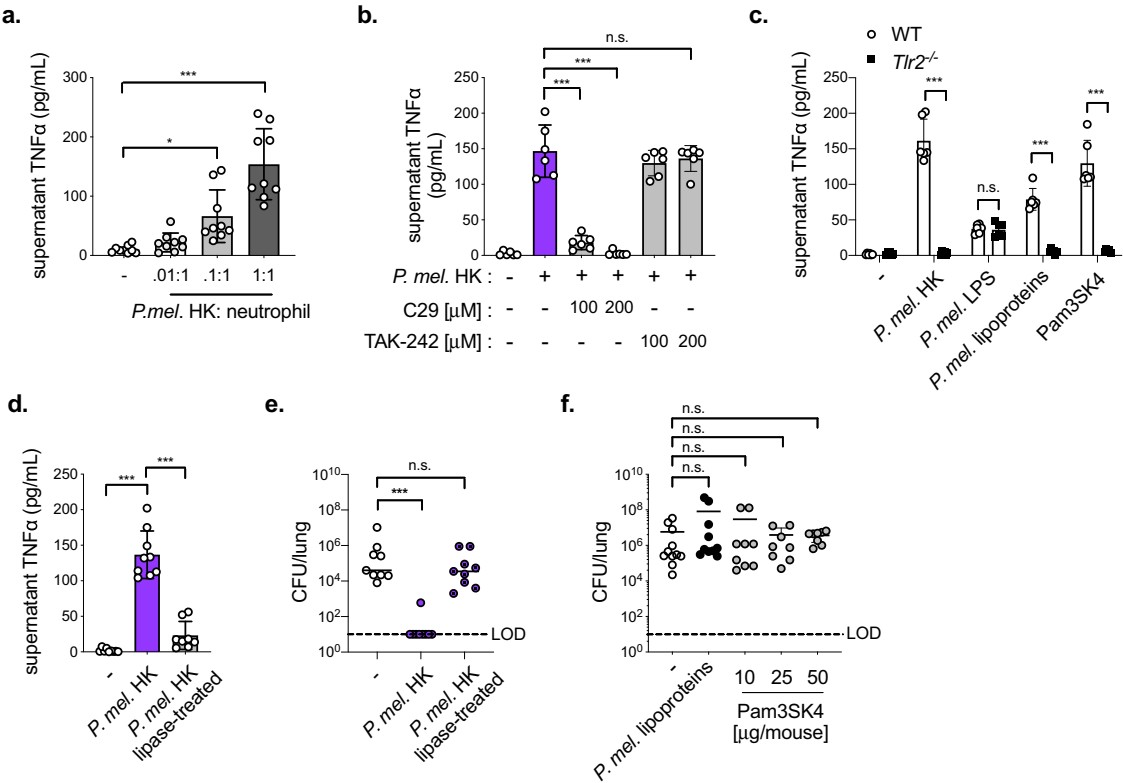

**Fig. 5 *P. melaninogenica* (*P. mel.*) lipoproteins induce neutrophil TNFα secretion in a TLR2-dependent manner. a** Supernatant TNFα detected 24 h following incubation of bone marrow (BM) neutrophils purified from naïve wild-type (WT) mice with *P. mel.* strain 25845 heat-killed (HK) at the indicated ratios ($n = 3$ independent experiments/group). **b** Supernatant TNFα detected 24 h following incubation of BM neutrophils from WT mice incubated with *P. mel.* HK (1:1 ratio) with and without inhibitors of TLR2 (C29) or TLR4 (TAK-242), ($n = 3$ independent experiments/group). **c** Supernatant TNFα detected 24 h following incubation of BM neutrophils from WT or *Tlr2*$^{-/-}$ mice with PBS (-), *P. mel.* HK (1:1 ratio), *P. mel.* lipopolysaccharide (LPS, 10 ng/mL), *P. mel.* lipoproteins (10 ng/mL), or Pam3SK4 (10 ng/mL), ($n = 3$ independent experiments/group). **d** Supernatant TNFα detected 24 h following incubation of BM neutrophils from WT mice treated with PBS (-), untreated *P. mel.* HK, or lipoprotein-digested *P. mel.* HK ($n = 3$ independent experiments/group). **e** Lung type 2 *S. pneumoniae* burdens in mice treated with PBS (-), untreated *P. mel.* HK, or lipoprotein-digested *P. mel.* HK i.t. prior to 24 h *S. pneumoniae* infection, $5 \times 10^6$ CFU/mouse ($n = 9$ mice/group). **f** Lung *S. pneumoniae* burdens in mice treated with PBS (-, $n = 11$ mice), *P. mel.* lipoproteins (10 μg/mouse, $n = 9$ mice), or Pam3SK4 (10 μg/mouse, $n = 10$ mice, 25 μg/mouse, $n = 8$ mice, or 50 μg/mouse, $n = 8$ mice) i.t. prior to 24 h *S. pneumoniae* infection. LOD = limit of detection. Data are pooled from three independent experiments, with cells plated in triplicate for in vitro assays. Data are displayed as mean ± SEM. For **a** *$p = 0.0107$, ***$p < 0.0001$, one-way ANOVA with Dunnett's post hoc test, **b** from left to right ***$p < 0.0001$, ***$p < 0.0001$, $p = 0.8011$, one-way ANOVA with Dunnett's post hoc test, **c** from left to right ***$p < 0.0001$, $p > 0.9999$, ***$p < 0.0001$, ***$p < 0.0001$, two-way ANOVA with Sidak's post hoc test, **d** ***$p < 0.0001$, one-way ANOVA with Tukey's post hoc test (**e**) from left to right ***$p = 0.0004$, $p = 0.1526$, Kruskal–Wallis with Dunn's post hoc test, **f** from left to right $p = 0.9344$, $p = 0.1490$, $p > 0.9999$, $p > 0.9999$, Kruskal–Wallis with Dunn's post hoc test. Source data are provided as a Source Data file.

TNFα was elevated following *P. melaninogenica* exposure (Fig. 8c). In contrast, WT mice exposed to *P. melaninogenica* had reduced serum TNFα and elevated IL-10 (Supplementary Fig. 7c). These results indicate that *P. melaninogenica*-induced IL-10 regulates systemic TNFα during *S. pneumoniae* infection and is important for *P. melaninogenica*-mediated protection. Analysis of lung myeloid cell TNFα expression revealed that TNFα was significantly elevated in several cell types, including neutrophils, inflammatory monocytes, and AMs, in *Il10*$^{-/-}$ mice, regardless of *P. melaninogenica* exposure (Fig. 8d–f and Supplementary Fig. 7d–e). Depletion of TNFα was not sufficient to reverse the loss of *P. melaninogenica*-mediated protection in *Il10*$^{-/-}$ mice (Supplementary Fig. 7f), suggesting a broad loss of IL-10-mediated restraint of myeloid cell activation. Together, these findings reveal that IL-10 regulation of lung inflammation is a critical component of *P. melaninogenica*-mediated protection against *S. pneumoniae* infection.

**Protection mediated by other airway *Prevotella*.** While *P. melaninogenica* is an abundant *Prevotella* species, it was unclear

whether other airway *Prevotella* isolates mediate a similar protective effect. As with *P. melaninogenica* strain 25845, used throughout these studies, exposure to live *P. melaninogenica* strain D18 significantly improved clearance of *S. pneumoniae* from the lung by 24 h post-infection (Fig. 9a). Three additional live airway *Prevotella* species, including *Prevotella buccae*, *Prevotella tannerae*, and *Prevotella nanceiensis*, also increased *S. pneumoniae* clearance from the lung (Fig. 9a). In contrast, the periodontal pathogen *P. intermedia* was not protective (Fig. 9a). These data suggest that several airway *Prevotella* species are capable of enhancing protection against *S. pneumoniae* infection. Finally, we compared the ability of HK preparations of each *Prevotella* species to activate neutrophils purified from the bone marrow of WT versus *Tlr2*$^{-/-}$ mice. Similar to *P. melaninogenica* strain 25845, strain D18 as well as *P. buccae*, *P. tannerae*, and *P. nanceiensis* induced neutrophil secretion of TNFα and IL-10 in a TLR2-dependent manner (Fig. 9b). In contrast, *P. intermedia* activated neutrophils in a TLR2-independent manner (Fig. 9b). These data are consistent with a critical role for TLR2-dependent neutrophil activation in species of *Prevotella* which are protective against *S. pneumoniae*.

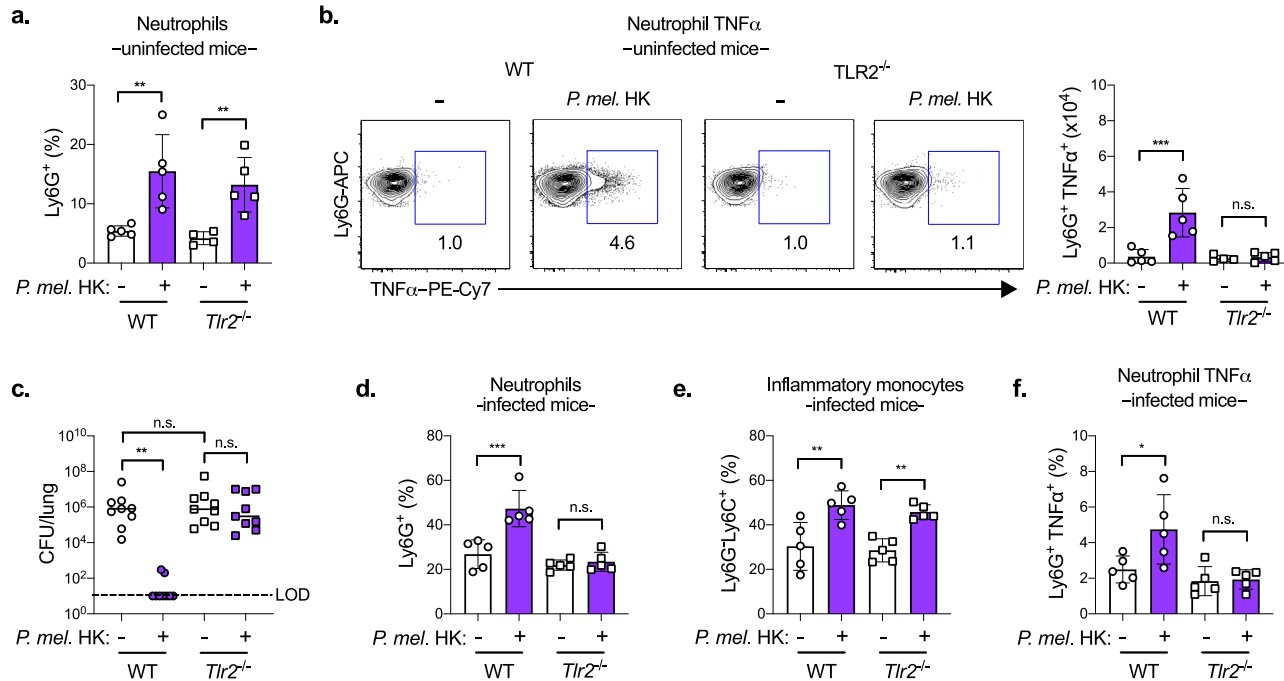

**Fig. 6 TLR2 is required for *P. melaninogenica* (*P. mel.*)-induced neutrophil TNFα and protection against *S. pneumoniae*. a, b** Percentage of neutrophils (**a**), representative flow cytometry plots and total cell numbers of neutrophil TNFα (**b**) detected by intracellular flow cytometry in WT or *Tlr2⁻/⁻* mice treated with either PBS (-) or *P. mel.* strain 25845 heat-killed (HK) i.t. for 24 h (*n* = 5 mice/group). **c–f** Lung type 2 *S. pneumoniae* burdens (*n* = 9 mice/ group) (**c**), percentage of neutrophils (*n* = 5 mice/group) (**d**), percentage of inflammatory monocytes (*n* = 5 mice/group) (**e**), and percentage of neutrophil TNFα (*n* = 5 mice/group) (**f**) detected by intracellular flow cytometry in mice treated with either PBS (-) or *P. melaninogenica* (*P. mel.*) HK i.t. prior to 24 h *S. pneumoniae* infection, 5 × 10⁶ CFU/mouse. LOD = limit of detection. Data are pooled from three independent experiments (**c**) or representative from one of four independent experiments (**a, b, d–f**), displayed as mean ± SEM. For **a** **p = 0.0026 (WT), **p = 0.009 (*Tlr2⁻/⁻*), one-way ANOVA with Sidak's post hoc test and p = 0.0338 (WT), p = .0081 (*Tlr2⁻/⁻*), Kruskal–Wallis with Dunn's post hoc test, **b** ***p = 0.0002 (WT), p = 0.9858 (*Tlr2⁻/⁻*), one-way ANOVA with Sidak's post hoc test and p = 0.0165 (WT), p > 0.9999 (*Tlr2⁻/⁻*), Kruskal–Wallis with Dunn's post hoc test, **c** from left to right **p = 0.0017, p > 0.9999, p > 0.9999, Kruskal–Wallis with Dunn's post hoc test, **d** *p = 0.0494 (WT), p > 0.9999 (*Tlr2⁻/⁻*), Kruskal–Wallis with Dunn's post hoc test, **e** **p = 0.0014 (WT), **p = 0.0026 (*Tlr2⁻/⁻*), one-way ANOVA with Sidak's post hoc test and p = 0.0162 (WT), p = 0.0400 (*Tlr2⁻/⁻*), Kruskal–Wallis with Dunn's post hoc test, **f** *p = 0.0142 (WT), p = 0.9898 (*Tlr2⁻/⁻*), one-way ANOVA with Sidak's post hoc test and p = 0.2056 (WT), p > 0.9999 (*Tlr2⁻/⁻*), Kruskal–Wallis with Dunn's post hoc test. Source data are provided as a Source Data file.

## Discussion

Work over the past decade has defined the gut-lung axis, highlighting a critical role for the gut microbiome in lung immune homeostasis. By comparison, our understanding of immune regulation by members of the airway microbiome is underdeveloped. Several individual members of the respiratory tract microbiome, including commensal *Streptococcus* and *Corynebacterium* species, produce factors that inhibit growth of *S. pneumoniae*[28–30]. However, it is less clear how these and other airway commensals regulate innate immune-mediated protection against lung infection[43]. Brown et al. showed that a compilation of Gram-positive airway bacteria including *Lactobacillus crispatus, Staphylococcus aureus*, and *Staphylococcus epidermidis* induced Nod2-dependent protection against *S. pneumoniae*[8]. In this case, a Nod2-activating bacterial cohort from the gut also induced protection, which required GM-CSF and AMs. As we did not detect a GM-CSF response to *Prevotella*, this suggests there are at least two distinct microbial-induced immune pathways capable of enhancing protection against *S. pneumoniae*. Sustained protection against *S. pneumoniae* infection in antibiotic treated and Germ-free mice indicates that *P. melaninogenica* is sufficient to drive immune-mediated protection without the participation of other members of the endogenous microbiota.

This study focused on the early response to *P. melaninogenica* in the context of acute lung infection. Recently, a compilation of oral commensals which included *P. melaninogenica* was shown to protect against *S. pneumoniae* for as long as 2 weeks[44]. Protection at this timepoint was associated with the induction of a Th17 response, which correlated primarily with the presence of *Veillonella* rather than *Prevotella*. However, RNA sequencing of lung tissue 24 h post-commensal aspiration confirmed the induction of TLR signaling and inflammatory responses including TNFα and IL-6, similar to our findings with *P. melaninogenica*. This indicates that exposure to either *P. melaninogenica* alone or in combination with other oral commensals enhances immune-mediated protection against *S. pneumoniae*. *Prevotella*-induced immune responses may coordinate with those induced by other commensal bacteria, such as *Veillonella*, for extended protection.

Inflammation itself is not automatically protective against *S. pneumoniae* infection. For example, virus-induced inflammation supports increased *S. pneumoniae* growth[45], and viral-pneumococcal co-infections are associated with higher morbidity and mortality[46]. Similarly, *S. pneumoniae* co-infection with *H. influenzae* involves synergistic inflammation, including the upregulation of TLR2 signaling in the lung and middle ear. *S. pneumoniae* is well-equipped for immune modulation and evasion, and expresses virulence factors that limit the efficacy of innate immune responses and impede neutrophil-mediated killing[47,48]. As a result, innate immune cells are typically insufficient for pneumococcal clearance. Our findings highlight a protective role for TLR2-dependent inflammation induced by airway *Prevotella* exposure, the timing of which may be important for overcoming *S. pneumoniae* immune evasion mechanisms. This may relate to reduced acquisition of *S. pneumoniae* in the

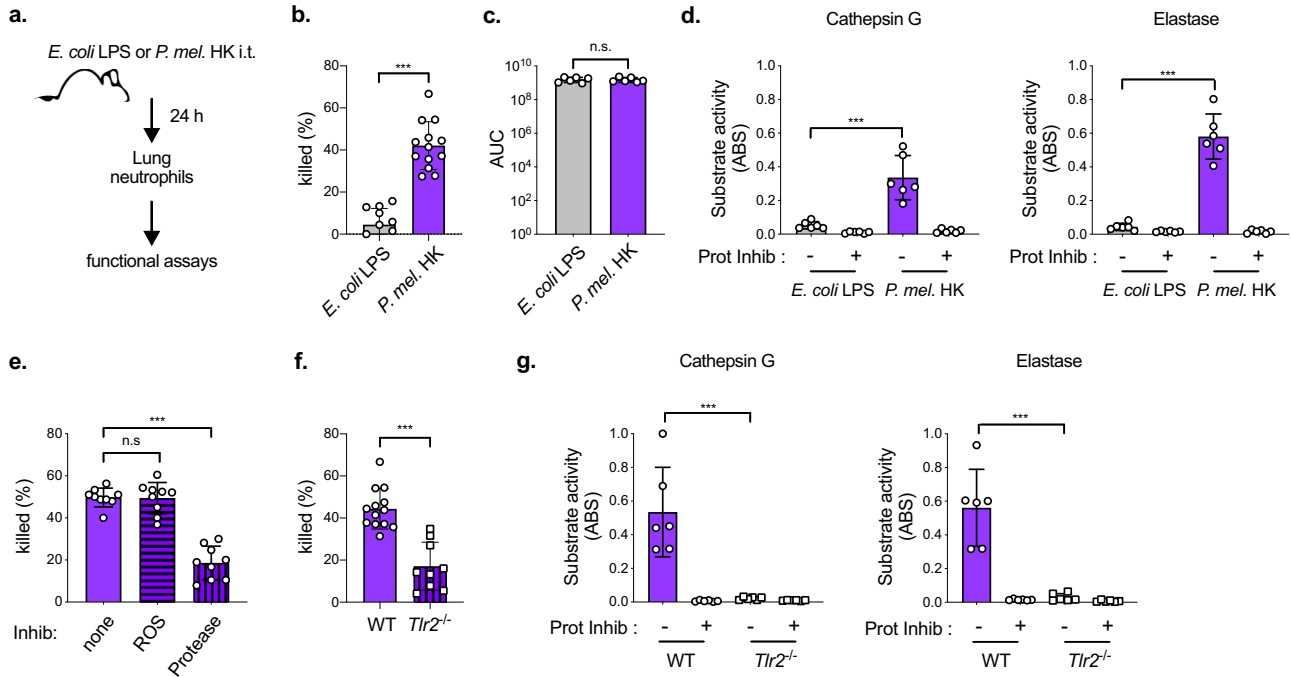

**Fig. 7 *P. melaninogenica* (*P. mel.*) enhances serine protease-mediated killing of *S. pneumoniae* by lung neutrophils in a TLR2-dependent manner.**
**a** Schematic of lung neutrophil purification. **b** Percent of type 2 *S. pneumoniae* killed by lung neutrophils purified from mice exposed to either *E. coli* lipopolysaccharide (LPS) or *P. mel.* strain 25845 heat-killed (HK) i.t. for 24 h following 1 h incubation with *S. pneumoniae* opsonized with 3% fresh mouse serum (n = cells isolated from 11 mice/group). **c** Total reactive oxygen species (ROS) produced in 1 h by lung neutrophils purified from mice exposed to either *E. coli* LPS or *P. mel.* HK i.t. for 24 h detected by luminol, with area under curve (AUC) shown (n = cells isolated from 6 mice/group). **d** Serine protease activity for cathepsin G and elastase +/− protease inhibitor cocktail (Prot Inhib) detected by substrate cleavage for lung neutrophils purified from WT mice exposed to either *E. coli* LPS or *P. mel.* HK (n = cells isolated from 6 mice/group). **e** Percent of *S. pneumoniae* killed by lung neutrophils purified from WT mice exposed to *P. mel.* HK i.t. for 24 h following 1 h incubation with opsonized *S. pneumoniae* in the presence of the ROS inhibitor DPI (ROS), protease inhibitors (Protease), or no inhibitors (none), (n = cells isolated from 9 mice/group). **f, g** Percent of *S. pneumoniae* killed by lung neutrophils purified from WT or *Tlr2*−/− mice exposed to *P. mel.* HK i.t. for 24 h (n = cells isolated from 13 WT mice, 9 *Tlr2*−/− mice) (**f**) or serine protease activity +/− protease inhibitor cocktail (Prot Inhib) (n = cells isolated from 6 mice/group) (**g**). Data are pooled from three independent experiments, with cells plated in duplicate or triplicate. Data are displayed as mean ± SEM. For **b** ***p < 0.0001, two-tailed t-test, **c** p = 0.9566, two-tailed t-test, **d** ***p = 0.0004 (Cathepsin G), ***p < 0.0001 (Elastase), two-tailed t-test, **e** p = 0.9926 (ROS), ***p < 0.0001 (Protease), one-way ANOVA with Dunnett's post hoc test, **f** ***p < 0.0001, two-tailed t-test, **g** ***p = 0.0008 (Cathepsin G), ***p = 0.0002 (Elastase), two-tailed t-test. Source data are provided as a Source Data file.

lungs of individuals with "optimal" innate immune priming by their airway microbiota.

The concept of trained immunity refers to non-specific beneficial immune priming, primarily mediated by innate immune signaling and directed by the microbiota. The critical cells and signaling pathways in the lung for trained immunity are not clear. Compared with all other tissues including the gut, the lungs have the highest expression of TLR2[49]. Immune priming by TLR agonists has been explored as a potential immunotherapy strategy for several infections, including *S. pneumoniae*. Specific combinations of TLR agonists are protective against *S. pneumoniae* infection, such as TLR2 together with TLR9 agonists[50]. *S. pneumoniae* lipoproteins are capable of activating TLR2[51], similar to *Prevotella*, but TLR2 is not sufficient for protection[52], as we confirm. One study, however, found that TLR2 is protective against pneumolysin-deficient *S. pneumoniae*, which no longer activates TLR4[53]. *Prevotella* express penta-acylated LPS, which can behave as a TLR4 agonist, in contrast to hexa-acylated LPS expressed by bacteria such as *E. coli*, which is a strong TLR4-inducer[54]. Further investigation is required to dissect the immune priming environment which coordinates with TLR2 to promote *P. melaninogenica*-mediated protection against *S. pneumoniae*, including the potential role of TLR4 antagonism. The protective capacity of TLR priming also differs by infection, as TLR4

priming with *E. coli*-derived LPS improves protection against lung infection with *Klebsiella pneumoniae* and influenza A virus[31,32], in contrast to *S. pneumoniae*. Together, these studies highlight the importance of understanding the context and mechanism of protective airway microbiome trained immunity.

While our data indicate an important role for neutrophils, it remains unclear which cells initiate neutrophil recruitment in response to *P. melaninogenica*. AMs are a likely candidate as the first immune cells to encounter inhaled particles and as a major source of neutrophil chemoattractants[33]. *P. melaninogenica* outer membrane vesicles induce AM responses including IL-17A and IL-17B[27], which can synergize with pro-inflammatory cytokines including TNFα. As others have shown[55], we find that TNFα is important for neutrophil recruitment during *S. pneumoniae* infection, and TNFα can prime activation of human neutrophils[56]. However, TNFα priming in neutrophils is typically associated with stronger NADPH oxidase activation and respiratory burst[57]. This contrasts with our findings of increased serine protease activity, which was dependent on additional signals in the *P. melaninogenica*-activated lung. Serine protease activity is essential for neutrophil killing of *S. pneumoniae*[58], and serine protease knockout mice have reduced clearance of *S. pneumoniae* from the lung[59], highlighting the importance of this pathway for infection control.

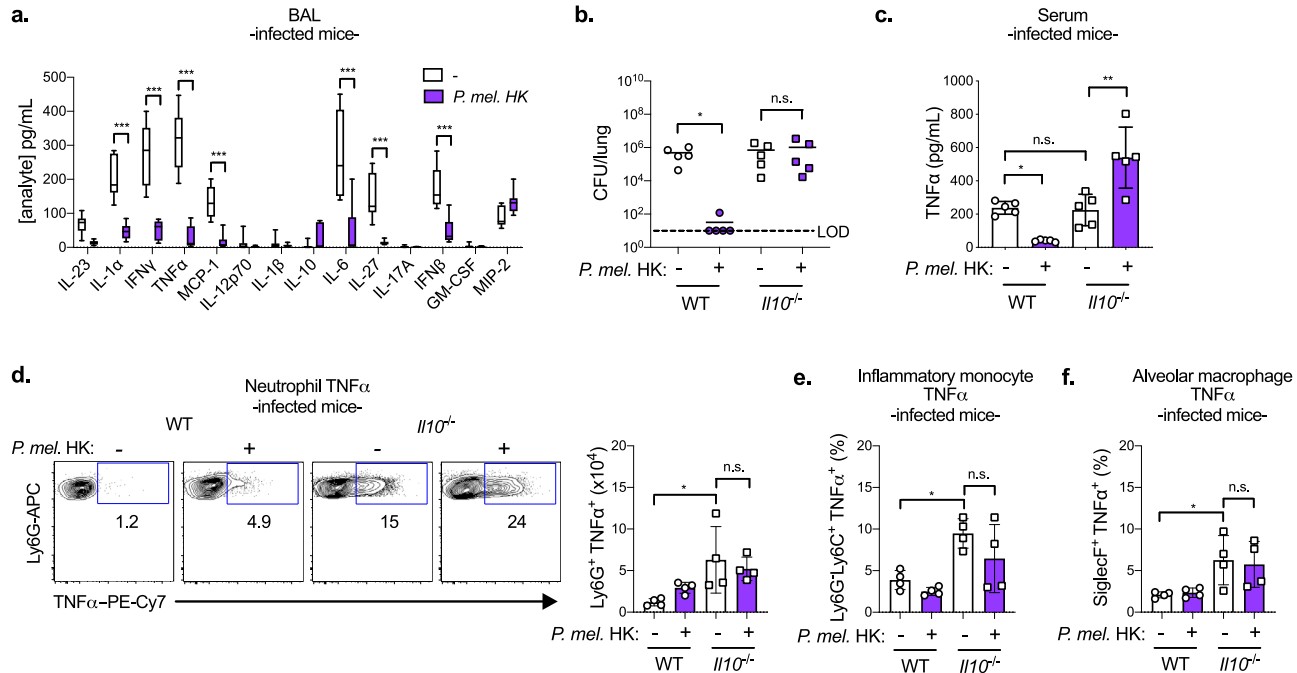

**Fig. 8 _P. melaninogenica_ (_P. mel._)-mediated protection against _S. pneumoniae_ requires restraint of detrimental inflammation by IL-10. a** Quantification of cytokines and chemokines in lung bronchoalveolar lavage (BAL) following exposure to PBS (-) or _P. mel._ strain 25845 heat-killed (HK) i.t. prior to 24 h type 2 _S. pneumoniae_ infection, $5 \times 10^6$ CFU/mouse ($n = 9$ mice/group). **b**, **c** Lung _S. pneumoniae_ burdens (**b**) and serum TNFα (**c**) detected in WT or _Il10^-/-_ mice exposed to either PBS (-) or _P. mel._ HK i.t. prior to 24 h _S. pneumoniae_ infection ($n = 5$ mice/group). **d–f** Representative flow cytometry plots and total cell numbers of neutrophil TNFα (**d**), percentage of inflammatory monocyte TNFα (**e**), and percentage of AM TNFα (**f**) detected by intracellular flow cytometry from WT or _Il10^-/-_ mice treated with either PBS (-) or _P. mel._ HK i.t. prior to 24 h _S. pneumoniae_ infection ($n = 4$ mice/group). LOD = limit of detection. Data are pooled from three independent experiments (**a**) or representative from one of four independent experiments (**b–f**). Box boundaries in (**a**) indicate the 25th and 75th percentiles, with a horizontal line representing the median and whiskers indicating minimum and maximum values. Data in **b–f** are displayed as mean ± SEM. For **a** ***$p < 0.0001$, two-way ANOVA with Sidak's post hoc test, **b** *$p = 0.0257$ (WT), $p > 0.9999$ (_Il10^-/-_), Kruskal–Wallis with Dunn's post hoc test, **c** from left to right *$p = 0.0393$, $p = 0.9973$, **$p = 0.0012$, one-way ANOVA with Tukey's post hoc test, **d** from left to right *$p = 0.0116$, $p = 0.7388$, one-way ANOVA with Sidak's post hoc test (**e**), from left to right *$p = 0.0101$, $p = 0.1698$, one-way ANOVA with Sidak's post hoc test, **f** from left to right *$p = 0.0280$, $p = 0.9298$, one-way ANOVA with Sidak's post hoc test. Source data are provided as a Source Data file.

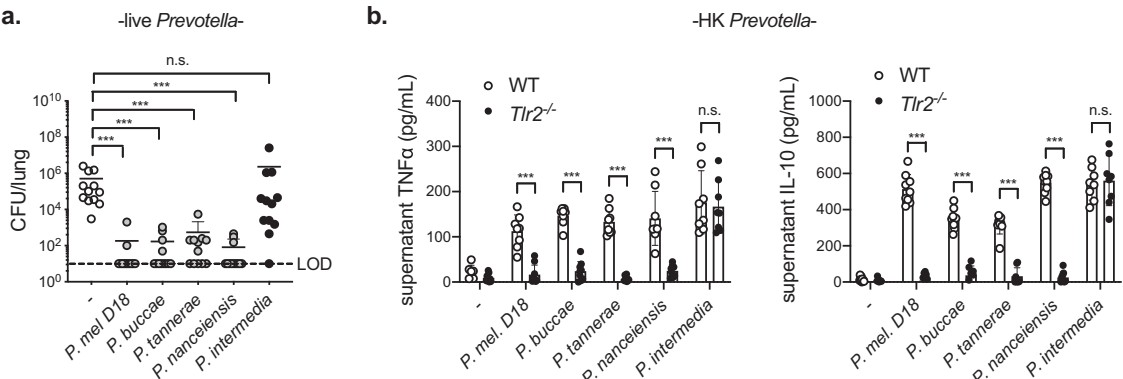

**Fig. 9 Protection against _S. pneumoniae_ is induced by other airway _Prevotella_ that activate neutrophils in a TLR2-dependent manner. a** Lung _S. pneumoniae_ burdens in mice treated with PBS (-), live _P. melaninogenica_ (_P. mel._) strain D18, live _P. buccae_, live _P. tannerae_, live _P. nanceiensis_, or live _P. intermedia_ i.t. prior to 24 h _S. pneumoniae_ infection, $5 \times 10^6$ CFU/mouse ($n = 12$ mice/group). **b** Supernatant cytokines detected 24 h following incubation of BM neutrophils with _P. mel._ strain D18 heat-killed (HK), _P. buccae_ HK, _P. tannerae_ HK, _P. nanceiensis_ HK, or _P. intermedia_ HK ($n = 3$ independent experiments/group). Data are pooled from three independent experiments, with cells plated in triplicate for in vitro studies, displayed as mean ± SEM. For **a** from left to right ***$p < 0.0001$, ***$p < 0.0001$, ***$p = 0.0005$, ***$p < 0.0001$, $p = 0.9999$, Kruskal–Wallis with Dunn's post hoc test, **b** from left to right, ***$p < 0.0001$, ***$p < 0.0001$, ***$p < 0.0001$, ***$p < 0.0001$, $p = 0.9959$ (TNFα), ***$p < 0.0001$, ***$p < 0.0001$, ***$p < 0.0001$, ***$p < 0.0001$, $p = 0.9948$ (IL-10), two-way ANOVA with Sidak's post hoc test. Source data are provided as a Source Data file.

Here, we find that *Prevotella*-mediated protection requires the co-induction of systemic IL-10, which regulates *Prevotella*-induced inflammation in *S. pneumoniae*-infected mice. In contrast to this beneficial role, we recently found that systemic IL-10 induced at 3 days post-infection exacerbates *S. pneumoniae* lung infection[60]. However, this response was delayed relative to the protective IL-10 response we observe in *Prevotella*-exposed mice. Others have shown that *P. melaninogenica* induces the production of IL-10 by human DCs[61], and we find that murine neutrophils secrete IL-10 in response to *P. melaninogenica*, suggesting that myeloid cell-derived IL-10 contributes to inflammatory control. These studies highlight the importance of inflammatory fine-tuning in the lung, where IL-10 plays a major role in limiting tissue damage, though at the risk of worsened infection under some circumstances.

Despite the genetic heterogeneity reported among *Prevotella* species[62], we identified two different isolates of *P. melaninogenica* and several other airway *Prevotella* species that enhance protection against *S. pneumoniae*. *P. melaninogenica* was previously shown to induce similar immune activation profiles, including the production of TNFα and IL-10 by lung CD45$^+$ cells, as *P. nanceiensis*[22], one of the strains we identify as protective. Although not as prevalent as *P. melaninogenica*, *P. buccae*, *P. tannerae*, and *P. nanceiensis* are also common in the upper and lower airway of healthy individuals[4,63]. In contrast, *P. intermedia*, which was not protective against *S. pneumoniae*, is a major periodontal pathogen. Others have reported that *P. intermedia* upregulates a critical adherence receptor used by *S. pneumoniae* in airway epithelial cells, exacerbating type 4 *S. pneumoniae* infection[25]. Here, we find that *P. intermedia* activates neutrophils in a TLR2-independent manner, unlike the other *Prevotella* strains, which are protective. The critical features of protective airway *Prevotella* species, including the role of TLR2-dependent neutrophil activation and regulation of the airway epithelium, is an important area for further investigation. Regardless, these findings suggest that the species-specific composition of airway *Prevotella* dictates infection outcome following *S. pneumoniae* exposure.

The immune activation profile we describe is consistent with several observations in humans, including the association between increased *Prevotella* and sub-clinical inflammation characterized by increased neutrophils, the human neutrophil chemotactic factor IL-8, Mip-1α, and IL-10 in the lungs of healthy adults[20,64]. Similarly, *Prevotella* abundance was associated with increased TNFα and IL-10 in the lungs of transplant patients, in whom *Prevotella* correlated with increased lung function and airway microbiome stability[19]. Finally, in vitro studies demonstrate the capacity of *P. melaninogenica* to induce TLR2 activation and responses including IL-6, TNFα, and IL-10 secretion by human monocytes and DCs[61,65,66]. Together, these findings indicate that *Prevotella* species including *P. melaninogenica* are associated with lung innate immune responses including TLR2 activation and the production of TNFα and IL-10 in both humans and animal models.

Our findings highlight an important role for airway *Prevotella* in directing immune-mediated protection in the respiratory tract. Further studies will be necessary to determine whether similar protective immune priming is induced by *Prevotella* aspiration in healthy adults and how such protection may be leveraged to reduce the burden of lung infections in susceptible individuals. Airway immunobiotics, or probiotics designed to enhance protective immunity, may be particularly impactful in individuals who are at increased risk of *S. pneumoniae* infection, though such use should be tempered by potentially detrimental effects, for example by exacerbating fibrosis in those with chronic lung disease. Alternatively, targeting the critical *Prevotella*-modified immune pathways could improve clearance of *S. pneumoniae* regardless of the presence of *Prevotella* as an immunotherapeutic strategy.

## Methods

**Animals.** Adult male and female mice aged 6-12 weeks of age were used for these studies. C57BL/6 J (WT), B6.129Tlr2tm1Kir (*Tlr2*$^{−/−}$), and B6.129il10tm1Cgn (*Il10*$^{−/−}$) mice were purchased from The Jackson Laboratory (stocks #000664, 004650, and 002251 respectively). All strains used in these studies (WT, *Tlr2*$^{−/−}$ and *Il10*$^{−/−}$) are on the C57BL/6J genetic background. Mice were maintained in the University of Colorado Office of Laboratory Animal Resources. Housing conditions included a light cycle of 14:10 (light:dark) hours, a temperature of 72 ± 2°F, with 40 ± 10% humidity. Mice were fed irradiated Tecklad diet (Envigo, Inotiv, Inc.), catalog #2920X for colony mice and #2919 for breeder pairs. Germ-free mice were obtained from the University of Colorado Anschutz Medical Campus Gnotobiotic Facility, which maintains a colony established with founder C57BL/6 mice obtained from the National Gnotobiotic Rodent Resource Center at the University of North Carolina. Germ-free mice are housed in sterilized vinyl film isolators with positive pressure air flow through HEPA filtration. Any items introduced into the isolators are sterilized, with quality control indicators to verify sterilization. The internal isolator environment and housed mice are tested bi-weekly and prior to experimental use for microbiota through culture-dependent methods and by qPCR (see Microbiome Depletion). For infection experiments, Germ-free mice were transferred directly from the Gnotobiotic Core Facility into BSL2 vivarium space. Transferred mice were exposed to input bacteria (see Infections) within 8 h of transfer.

**Microbiome depletion.** Antibiotic treated mice were exposed to a broad-spectrum antibiotic cocktail (ampicillin 1 g/L, neomycin 1 g/L, metronidazole 1 g/L, vancomycin 0.5 g/L, MilliporeSigma and McKesson) in drinking water ad libitum for 7 days. Water containing antibiotics was replaced with normal drinking water 48 h prior to live *Prevotella* exposure. Microbiome depletion was confirmed by qPCR using genomic DNA extracted from stool samples using the PureLink™ Genomic DNA Mini Kit (ThermoFisher Scientific). Primers (ACTCCTACGGGAGGCAG-CAGT and ATTACCGCGGCTGCTGGC) were used with iTaq™ Universal SYBR® Green Supermix (BioRad) and 1 μL template DNA, with reactions performed on a CFX Connect™ Real-Time System (BioRad) under the following cycle conditions: (1) 94 °C 4 min; (2) 40 cycles of 15 s at 95 °C, 30 s at 60 °C, and (3) 72 °C for 10 min. Total 16 S rRNA gene copy numbers were calculated using a standard curve generated with a known concentration of *S. pneumoniae* D39 gDNA, input ng/μL DNA, and Ct values. qPCR data were analyzed using CFX Manager Software (version 2.1, BioRad).

**Bacteria.** *Prevotella melaninogenica* strain ATCC® 25845™, *Prevotella tannerae* strain ATCC® 51259™, and *Prevotella intermedia* strain ATCC® 25611™ were obtained from the American Type Culture Collection, Manassas VA. *Prevotella melaninogenica* strain D18 (catalog #HM-80) and *Prevotella buccae* strain D17 (catalog #HM-45) were obtained from BEI resources, NIAID, NIH as part of the Human Microbiome project. *Prevotella nanceiensis* strain PP1746 was a kind gift from Dr. Paul Planet, Division of Infectious Diseases at the Children's Hospital of Philadelphia. *Prevotella* strains were grown under anaerobic conditions on Brucella Agar containing 5% sheep blood, hemin and vitamin K at 37 °C for 72 h (ThermoFisher Scientific). Heat-killed *Prevotella* were prepared from fresh plates following resuspension in PBS and incubation at 56 °C for 35 min. Samples before and after heat-killing were used to determine CFU equivalents/mL and confirm killing, respectively. Heat-killed *Corynebacterium accolens* strain ATCC® 49726™ (American Type Culture Collection) and *Corynebacterium amycolatum* strain SK46, catalog #HM-109 (BEI resources, NIAID, NIH as part of the Human Microbiome project) were prepared following growth in BHI broth cultures supplemented with 1% Tween® 80 (polysorbate, VWR). Heat-killed *Streptococcus salivarius* strain SK126, catalog #HM-109 (obtained from BEI Resources, NIAID, NIH as part of the Human Microbiome project) was prepared following growth in Todd Hewitt Broth with 5% Yeast Extract (BD Bacto™) at 37 °C with 5% CO$_2$ without shaking. Heat-killed *Escherichia coli* (strain DH5α, ThermoFisher Scientific) was prepared following growth in LB broth (BD Bacto™) at 37 °C with shaking (200 rpm). A streptomycin resistant variant of serotype 2 *Streptococcus pneumoniae* strain D39 was a kind gift from Dr. Jeffrey N. Weiser (New York University). Serotype 3 *S. pneumoniae* strain ATCC® 6303™ (American Type Culture Collection) was also used. *S. pneumoniae* was grown in Todd Hewitt Broth with 5% Yeast Extract (BD Bacto™), with 50 μg/mL streptomycin (Sigma), added for type 2 strain D39 only, at 37 °C with 5% CO$_2$ without shaking.

**Infections.** For live *P. melaninogenica* infections, bacterial suspensions from fresh plates were prepared in PBS to an optical density (OD$_{600}$) of 0.3, centrifuged at ≥20,000 x g for 10 min, and resuspended in PBS prior to injection. *S. pneumoniae* was grown in broth from frozen stocks to mid-log phase and centrifuged at ≥20,000 x g for 10 min followed by resuspension in PBS for infections. Inoculum burdens were determined by serial dilution for CFU enumeration. Lungs and spleens collected from infected mice were homogenized using a Bullet Blender

tissue homogenizer (Stellar Scientific, Baltimore, MD). Tissue burdens were calculated following serial dilution in PBS and growth on Tryptic Soy agar plates containing neomycin (5 μg/mL, Sigma) and streptomycin (50 μg/mL, D39$^{STR}$ strain only) prepared with fresh catalase (5000 Units/plate, Worthington Biochemical Corporation, Lakewood, NJ). Plates were grown at 37 °C with 5% $CO_2$ for 24 h. All intratracheal and intranasal infections were conducted in a volume of 50 μL on mice anesthetized by inhaled isoflurane. For cell and cytokine depletions, mice were treated intraperitoneally 24 h prior to *S. pneumoniae* infection with 200 μg/mouse isotype control anti-IgG2A antibody (clone C1.18, catalog #BE0085, lot #722719J2), anti-Ly6G antibody (clone 1A8, catalog #BE0071-1, lot #80772101), or anti-TNFα antibody (clone XT3.11, catalog #BE0058, lot #728221A1) as indicated (Bio X Cell, Lebanon, NH). For *E. coli* LPS (0111:B4, Sigma), *P. melaninogenica* lipoprotein (described below) and Pam3SK4 treatments, mice were treated with 10 μg (LPS) or 10–50 μg (Pam3SK4) i.t. 24 h prior to *S. pneumoniae* infection. For survival assays, moribund mice humanely euthanized.

**Flow cytometry**. Lungs were harvested following perfusion by transcardial injection of 10 mL PBS, and single cells were prepared for flow cytometry[60]. Briefly, lungs were subjected to mechanical (mincing) and enzymatic (DNAseI 30 μg/mL, Sigma, and type 4 collagenase 1 mg/mL, Worthington Biochemical Corporation) digestion prior to passage through a 70 μM strainer. Red blood cells were lysed in RBC lysis buffer (0.15 M $NH_4Cl$, 10 mM $KHCO_3$, 0.1 mM $Na_2EDTA$, pH 7.4). Fc receptors were blocked by incubation in anti-CD16/32 (2.4G2 hybridoma supernatant) prior to staining in FACS buffer (1% BSA, 0.01% NaN3, PBS). For intracellular flow cytometry, cells were incubated with Brefeldin A (BD Biosciences) prior to staining and permeabilized with 1 mg/mL saponin (Sigma) prior to intracellular staining. All cells were fixed in 1% paraformaldehyde. Antibodies used for staining included anti-mouse: Siglec F (BD, catalog #562681, clone E50-2440, lot #B302914), MHCII (BioLegend, catalog #107643, clone M5/114.15.2, lot #B317262), Ly6G (BioLegend, catalog #127614, clone 1A8, lot #B292772), Ly6C (BioLegend, catalog #128012, clone HK1.4, lot #B250462), CD45.2 (BD, catalog #564616, clone 104, lot #1083734), CD11c (BioLegend, catalog #117338, clone N418, lot #B290360), CD11b (BioLegend, catalog #101212, clone M1/70, lot #B281906), and TNFα (ThermoFisher Scientific, catalog #25-7321-82, clone MP6-XT22, lot #2044683). All antibodies were used at a 1:200 dilution for staining. Flow cytometry was performed on an LSR Fortessa X-20 in the ImmunoMicro Flow Cytometry Shared Resource Laboratory at the University of Colorado Anschutz Medical Campus (RRID:SCR_021321). Data analysis was performed using FlowJo™ Software, version 9.9.6 (BD Life Sciences).

**Cytokine and chemokine analysis**. BAL cytokines and chemokines with the exception of MIP-2 were measured using a LEGENDplex™ Mouse Inflammation Panel (BioLegend), with analytes detected on the LSR Fortessa X-20 in the ImmunoMicro Flow Cytometry Shared Resource Laboratory at the University of Colorado Anschutz Medical Campus (RRID:SCR_021321). Data were analyzed using the LEGENDplex™ Data Analysis Software Suite (BioLegend). BAL MIP-2 was measured using a mouse CXCL2/MIP-2 ELISA kit (R&D Systems), serum cytokines were measured using mouse IL-10 and TNFα ELISA kits (BD), and analytes were detected on a Synergy™ HT Microplate Reader (BioTek). Data were analyzed using Prism (GraphPad, version 8).

**Cell isolations and stimulations**. Bone marrow neutrophils were isolated from the femurs of mice by Histopaque density gradient centrifugation[67] and purity was confirmed to be >80% Ly6G$^+$ neutrophils by flow cytometry. *P. melaninogenica* lipoproteins were prepared by Triton X-114 phase partitioning[68] and *P. melaninogenica* LPS was prepared using an LPS isolation kit (Sigma). Concentrations were determined relative to Pam3SK4 and *E. coli* LPS standard curves by running purified lipoprotein and LPS preparations on a 15% sodium dodecyl-sulfate polyacrylamide gel electrophoresis gel prior to detection using a Silver Stain Kit (Bio-Rad Laboratories, Inc) and imaging on a ChemiDoc XRS + Gel Imaging System (Bio-Rad Laboratories, Inc). *P. melaninogenica* LPS endotoxin activity was confirmed using a Pierce™ Chromogenic Endotoxin Quant Kit (ThermoFisher Scientific). Lipoprotein lipase-treated *P. melaninogenica* HK was prepared by incubation with 200 μg lipoprotein lipase (Sigma) for 2 h at 37 °C. For cell stimulation assays, 10$^5$ neutrophils in RP10 media supplemented with penicillin and streptomycin were exposed to *P. melaninogenica* HK (1:1 ratio), lipase-treated *P. melaninogenica* HK, *P. melaninogenica* lipoprotein (10 ng/mL), *P. melaninogenica* LPS (10 ng/mL), C29 TLR2 inhibitor (100–200 μM, Selleck Chemicals), Resatorvid TAK-242 TLR4 inhibitor (100–200 μM, Selleck Chemicals), or Pam3SK4 (10 ng/mL, InvivoGen, San Diego, CA) and incubated for 24 h at 37 °C with 5% $CO_2$ prior to supernatant collection. For cell infection assays, 10$^5$ neutrophils were exposed to 10$^5$ *S. pneumoniae* for 1 h prior to washing and incubation for 24 h at 37 °C with 5% $CO_2$ with or without *P. melaninogenica* HK (1:1 ratio) in media containing gentamycin (10 μg/mL).

**Neutrophil functional assays**. Lung neutrophils were isolated by positive selection (MojoSort PE-positive selection kit, BioLegend), and purity was confirmed to be >90% Ly6G$^+$ neutrophils by flow cytometry. Similar results were obtained using lung neutrophils isolated by negative isolation (19762, Mouse Neutrophil

Enrichment kit, STEMCELL Technologies). Neutrophils were isolated from the lungs of mice following 24 h treatment with PBS, *E. coli* LPS, or *P. melaninogenica* HK i.t. as indicated. For opsonophagocytic killing assays, 10$^3$ *S. pneumoniae* grown to mid-log phase were opsonized for 30 min with 3% fresh mouse sera prior to incubation with 10$^5$ neutrophils in Hank's buffer/0.1% gelatin for 1 h at 37 °C under rotation. Killing assays were completed in the presence or absence of 1x Halt™ protease inhibitor cocktail (ThermoFisher Scientific) or 10 μM diphenyle-neiodonium chloride (DPI), (ThermoFisher Scientific), which we found did not impact neutrophil viability over the course of 1 h. CFUs were measured by serial dilution plating, and percent killing was determined relative to reactions without neutrophils. Serine protease activity was determined using substrates specific to elastase (0.85 mM MeOSuc-Ala-Ala-Pro-Val-*p*NA, Sigma) and cathepsin G (0.1 mM Succinyl-Ala-Ala-Pro-Phe-pNA, Sigma)[58]. Briefly, 10$^5$ purified neutrophils were incubated with or without 1x Halt™ protease inhibitor cocktail for 30 min prior to washing and lysis in 0.1% Triton X-100. Substrates were added to neutrophil lysates and incubated in the dark for 45 min at 37 °C. Absorbance was determined by reading the OD$_{410}$ on Synergy™ HT Microplate Reader (BioTek) minus control wells with no neutrophils added. For detection of total ROS, 10$^5$ purified neutrophils were resuspended in KRP buffer (5 mM glucose, 1 mM $CaCl_2$, 1 mM $MgSO_4$ in PBS) and equilibrated for 15 min prior to plating on Greiner Bio-One LUMITRAC™ plates (FisherScientific) followed by the addition of 50 μM luminol (FisherScientific)[69]. Luminescence over 1 h at 37 °C was detected using a Synergy™ HT Microplate Reader (BioTek). Area under curve (AUC) was calculated, minus wells with no neutrophils added.

**Study approval**. These studies were approved by the Animal Care and Use Committee of the University of Colorado School of Medicine (protocol #00927) and by the Institutional Biosafety Committee (protocol #1418).

**Statistical analysis**. All measurements were taken from distinct samples, with exact sample sizes indicated in Figure legends. Prism (GraphPad, version 8) was used for all statistical analysis. All data were tested for normality using the Shapiro–Wilk test. For data with normal distributions, two-tailed Student's *t*-tests, one-way or two-way ANOVA tests with Dunnett's, Sidak's, or Tukey's post hoc analyses for multiple comparisons were used as specified. Two-tailed Mann–Whitney $U$-tests and Kruskal–Wallis tests with Dunn's post hoc analysis for multiple comparisons were used for all data with non-Gaussian distributions. Log-rank Mantel-Cox test was used for survival group comparison. For all statistical tests, $p$-values < 0.05 was considered significant. Individual $p$-values are specified in Figure Legends.

**Reporting summary**. Further information on research design is available in the Nature Research Reporting Summary linked to this article.

## Data availability

All data supporting the findings of this study are available within the paper (and its supplementary files). Source data are provided with this paper.

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

## Acknowledgements

The authors would like to acknowledge their colleagues in the Department of Otolaryngology and Department of Immunology and Microbiology for critical discussion of these data. Research reported in this publication was supported by the National Institute of Allergy and Infectious Diseases of the National Institutes of Health under Award Number K22AI143922 (S.E.C.), an American Thoracic Society Research Program Grant (S.E.C.), an American Lung Association Innovation Award (S.E.C.), and a Boettcher Foundation Webb-Waring Biomedical Research Award (S.E.C.).

## Author contributions

S.E.C. conceived and designed the experiments and wrote the original manuscript draft. S.E.C., K.J.H., and M.A.S. performed the experiments, analyzed data, and edited the manuscript draft. Z.G.D. performed experiments and analyzed data.

## Competing interests

The authors declare no competing interests.
