## [Peer Review File · Nature Communications]

REVIEWER COMMENTS

Reviewer #1 (Remarks to the Author):

This manuscript, "Airway *Prevotella* promote TLR2-dependent neutrophil activation and rapid clearance of *Streptococcus pneumoniae* from the lung", provides an interesting addition to how resident microbes in the airway help protect against pathogenic microbes. It details how *Prevotella melaninogenica* (Pm) protects against *Streptococcus pneumoniae* (Sp) lung infection. Overall, I found the study to be thorough and well controlled. I think it adds several important findings to the field. I have the following points:

- 1) There are a few additional controls which would support the authors' mechanism. One confounder not addressed here is the indirect effect of Pm inoculation on microbial communities already resident in mice. Could the effect of Pm be indirect through its modulation of abundance of other organisms already resident in the upper airway or lung? This could be addressed by using antibiotic treatment in their existing models.
- 2) Could any of the effects be driven from the gut? More specifically, are any of the Pm inoculated via the different routes used in this study getting to the gut and signalling from there?
- 3) Does this effect extend to other *Prevotella*? The authors only tested a very limited repertoire of other commensals.
- 4) The analysis of the TNF/neutrophil axis is good. I think it's important to understand what the TNF levels in the lung are during neutrophil depletion. This would help to understand whether neutrophils are the sole lung TNF source or not.
- 5) In the IL-10 KO model, if TNF is neutralized does this rescue the defect caused by IL-10 loss?
- 6) In the experiments where synthetic TLR2 ligands are administered to mice, they fail to promote Sp clearance promoting the authors to suggest TLR2 is necessary but not sufficient. However, could this be simply a dosing issue? Has a comparable amount of synthetic TLR2 ligand been added to the level of TLR2 agonists produced/displayed by Pm?
- 7) Are the serine protease inhibitors directly toxic to Sp?
- 8) One of the most striking findings is that it looks like TLR2 is not important for direct recognition of Sp. Care has to be taken with the TLR2^{-/-} mice because direct recognition of Sp could be lost (in addition to recognition of Pm). As a minimum, there must be some statistical comparison in the data shown that confirms, at the point of infection shown in most of the data (24h) TLR2 is not important for Sp defence and clearance.
- 9) Tlr2^{-/-} should be used instead of TLR2^{-/-} for mouse data.

Reviewer #2 (Remarks to the Author):

Review Horn et al - Nat Communications Jan 2022

Airway Prevotella promote TLR2-dependent neutrophil activation and rapid clearance of *Streptococcus pneumoniae* from the lung

Kadi J. Horn, Melissa A. Schopper, Sarah E. Clark*

This study investigates the potential interaction of *Prevotella melaninogenica*, one of the most commonly isolated species of the lung microbiome, with the bacterial pathogen *Streptococcus pneumoniae*, the most common cause of community-acquired pneumonia.

A clear point is made on associations between the presence of *Prevotella* spp. and the induction of a milder inflammation (sub-clinical with neutrophil involvement) in the context of different pathogenic infections. Microbiome analyses have indicated that increased abundance of *P. melaninogenica* correlates with reduced infections.

Using a mouse co-infection model, the presented data show that *P. melaninogenica* induced an innate immune that resulted in some immune protection against *S. pneumoniae* infection through recognition via TLR2, induction of TNF α , and neutrophil activation. Increased clearance of *S. pneumoniae* from the lung was due to increased serine protease-mediated killing resulting in improved survival.

The results presented here are highlighting the interactions of co-infections in the activation of the immune system. While a suggestion can be made regarding the importance of the whole microbiome in this process, as the study focused on a co-infection model consisting of two bacterial species, this should be made clear in the abstract and the conclusion.

This would take nothing away from the importance of the data but highlights that with our current methods we are not yet able to investigate the whole microbiome interactions in this manner.

Overall, the presented work is highly relevant to respiratory health and emphasizes the importance of co-infections (the microbiome?). Experiments are thoroughly performed, however, there are too

many graphs (some show duplicate results). The manuscript would also benefit from some cleared and shorter description of the results.

Methods:

Various bacterial strains were used throughout the study. Most of them were grown in their specific medium or broth. Would the medium the strain was propagated in have any effect on its virulence?

Please comment.

Animals: Please confirm that there is no potential mispairing of C57BL/6 sub-strains (here B6.129Tlr2tm1Kir (TLR2^{-/-}), and B6.129il10tm1Cgn (IL-10^{-/-} 4) and the C57BL/6J (WT) wild-type strain.

Please comment and add into the manuscript.

Prevotella aspiration was modelled in mice by instilling *P. melaninogenica* intratracheally (i.t.) prior to challenge with *S. pneumoniae*. The overall rationale for this was that the murine airway microbiome contains both *Prevotella* and *Streptococcus* species, but *P. melaninogenica* and *S. pneumoniae* are not resident members.

While the model allows for the addition of the required species to the existing lung microbiome, how does such addition change the whole microbiome (not only the response to each other, inflammatory immune responses, or bacterial clearance)? Can the overall effect be attributed to the interaction of *P. melaninogenica* with *S. pneumoniae* or with the overall change in the microbiome?

Please comment and add the information to the manuscript.

Statistical analysis: Experiments are performed at n=4-5 in each experiment.

Although the plotted data in the dot plot look as if they are normally distributed, the authors should consider non-parametric tests and descriptive measures for the small numbers investigated.

Results:

Pre-exposure (instillation) of heat-killed *P. melaninogenica* increased survival following a lethal dose of *S. pneumoniae* (fig 1b), however, when equivalent doses of *P. melaninogenica* and of *S. pneumoniae* were used only heat killed bacteria mediated the protection.

“Heat-killed *P. melaninogenica* was protective against both serotype 2 *S. pneumoniae*, which spread systemically by 24 hours, and serotype 3 *S. pneumoniae*, which was restricted to the lung (Fig. 1d-e)”

□ Early clearance of a sublethal dose of *S. pneumoniae* in mice exposed to *P. melaninogenica* correlated with lower burdens in the lung 10 three days later (Supplementary Fig. 1a)

In subsequent experiments only heat-killed *P. melaninogenica* was used. Importantly, heat killed *E. coli* did not mediate protection (Fig. 1f). However, as both bacteria are Gram negative species and TLR4 agonists, it appears that not all Gram-negative bacteria have a similar effect

Analyses of serum cytokines indicated that the protective effect heat-killed *P. melaninogenica* was mediated by the induction of IL-10 and a reduction of TNF α , differences in cytokines only seen in *P. melaninogenica* exposures.

What is the difference between Fig. 1g (“... less TNF α and more IL-10 at 24 hours post-*S. pneumoniae* infection compared to those exposed to *E. coli* HK or *E. coli* LPS ...”) and Fig. 2b (“Mice exposed to *P. melaninogenica* also had increased systemic TNF α and IL-10 compared to mice treated with PBS or *E. coli* LPS”). Can these graphs be combined?

Further analyses of BAL fluid showed a panel of innate immune cytokines (e.g., increase in IL-1 β , IFN γ , MCP1, IL6, MIP-2) and the recruitment of neutrophils to the lungs after *P. melaninogenica* exposure (Ly6G⁺ cells). In contrast to *E. Coli*, *P. melaninogenica* also showed a stronger activation of the neutrophils (Ly6G⁺ / TNF α ⁺ cells).

Monocytes were recruited to the lungs by both strains (Ly6G⁻ / Ly6C⁺ cells) and activated (Ly6G⁻ / Ly6C⁺ / TNF α ⁺ cells).

This overall suggests a “selective’ effect of *P. melaninogenica* on neutrophils, which potentially mediates the immune protection. When Ly6G⁺ cells or TNF α ⁺ from Ly6G⁺ cells are deleted, the protective effect on bacterial burden ceases.

Investigations into the host and bacterial requirements for *P. melaninogenica*-induced neutrophil TNF α production showed the importance of TLR2 signalling for this response (inhibition of TLRs). Further, TNF α induction was significantly reduced when heat-killed *P. melaninogenica* was treated with lipase to digest bacterial lipoproteins in the cell membrane, which are bacterial TLR2 ligands.

However, of interest is that additional factors appear to be required to mediate the protective TLR2-lipoprotein signalling effect as neither Pam3SK4 (TLR2 agonist) nor purified *P. melaninogenica* lipoproteins reduced *S. pneumoniae* burdens in the lung (Fig. 4f).

TLR2^{-/-} mice showed that TLR2 is required for *P. melaninogenica*-induced neutrophilic activation and TNF α secretion.

In the co-infection model, neutrophil recruitment was lost in *P. melaninogenica*-exposed TLR2^{-/-} mice infected with *S. pneumoniae*, while the recruitment of inflammatory monocytes was maintained.

This is an important finding that should be added to the figures.

The effect of IL10 in the regulation of protective effects of TNF α

Using purified neutrophils, *P. melaninogenica* induced the secretion of IL-10 in a dose-dependent manner, which could regulate TNF α production. Compared to the single infections, co-infection of bone marrow derived neutrophils with *P. melaninogenica* and *S. pneumoniae* shows significantly reduced TNF α release. This is also shown as a systemic response in *Prevotella*-exposed mice.

This is not in Fig. 1g, but in fig 7?

Additionally, BALF cytokines TNF α , IL-6, IL-1 β , IFN γ and IFN γ were also reduced in co-infections. This was observed 48h after co-infections, suggesting that a regulator such as IL-10 must be induced, transcribed etc. before an effect can be seen on protein levels.

IL-10 deficient (IL-10 $^{-/-}$) animals

Unexpectedly, *P. melaninogenica*-mediated protection against *S. pneumoniae* was lost in IL-10 17 $^{-/-}$ mice, which had similarly high burdens regardless of *P. melaninogenica* exposure (Fig. 7b). . In IL-10 18 $^{-/-}$ mice, serum TNF α was elevated following *P. 19 melaninogenica* exposure, rather than reduced as in WT mice (Fig. 7c). Further, WT mice 20 exposed to *P. melaninogenica* had significantly elevated systemic IL-10 compared to mice 21 infected with *S. pneumoniae* alone (Fig. 7c).

Additional effect of IL-10 derived from other immune cells.

Detailed comments

Line 5, page 2: “*Prevotella* are frequently [identified / found] among the top three most abundant bacteria detected in the oral cavity and lungs of healthy adults” – add verb

Line 26, page 5 ff. “... which like *S. pneumoniae* are gram-positive.” Please refer to Gram staining as Gram-positive or Gram-negative, acknowledging Dr Gram, the Danish bacteriologist who developed the technique in 1884. (<https://www.newscientist.com/article/2216418-hans-christian-gram-the-biologist-who-helped-investigate-bacteria/>). Please correct throughout.

Line 14, page 20: “Ly6G antibody (clone 1A8), or ant-TNF α antibody (clone XT3.11) as indicated (Bio X Cell ...” – add ‘i’ (anti).

Line 12, Page 21: “ ... with analytes were detected on the 14 LSR Fortessa X-20 in the ImmunoMicro Flow Cytometry Shared Resource Laboratory ...” Omit ‘were’

Please check for typographical and grammatical errors throughout.

Response to Referees

April 28th, 2022

Manuscript: NCOMMS-21-49627A

Title: **Airway *Prevotella* promote TLR2-dependent neutrophil activation and rapid clearance of *Streptococcus pneumoniae* from the lung**

Summary of Changes: We thank the reviewers for their time and critical analysis of our manuscript. Our revised manuscript includes extensive new data to address reviewer comments. Among these new data, we have included a new Figure and Supplementary Figure [Fig. 4, Supplementary Fig. 4] focused on the role of the endogenous microbiome in *P. melaninogenica*-mediated protection against *S. pneumoniae* infection. We find that live *P. melaninogenica* is protective against *S. pneumoniae* in both antibiotic treated and in Germ-free mice, suggesting that exposure to *P. melaninogenica* is sufficient to enhance *S. pneumoniae* clearance without the participation of the endogenous microbiota [Fig. 4a-b]. In Germ-free mice, we confirm that *P. melaninogenica* induces neutrophil recruitment and activation in the lungs of infected mice [Fig. 4c], similar to mice with an intact microbiome. We also address whether other airway *Prevotella* mediate a similar protective effect against *S. pneumoniae* infection in a new Figure [Fig. 9]. We find that exposure to several other live airway *Prevotella* isolates, including another strain of *P. melaninogenica* (D18), *P. buccae*, *P. tannerae*, and *P. nanceiensis*, mediates rapid clearance of *S. pneumoniae* from the lung, similar to the effect of *P. melaninogenica* strain 25845 used throughout the original study. In contrast, the major periodontal pathogen *P. intermedia* does not improve protection against *S. pneumoniae* [Fig. 9a]. In addition, we find that only *Prevotella* species which are protective against *S. pneumoniae* induce neutrophil secretion of TNF-alpha and IL-10 in a TLR2-dependent manner [Fig. 9b]. Together, these data are consistent with an important role for TLR2-dependent neutrophil activation in *Prevotella*-mediated protection against *S. pneumoniae*. Additional new data [Fig. 5f, Supplementary Fig. 2h, Supplementary Fig. 6c, Supplementary Fig. 7a, f] and highlighted edits to the manuscript text are described in the point by point responses below. We believe these new data prompted by reviewer input significantly improve the strength of our revised manuscript.

Responses to individual referees:

Reviewer #1 (Remarks to Author):

This manuscript, "Airway *Prevotella* promote TLR2-dependent neutrophil activation and rapid clearance of *Streptococcus pneumoniae* from the lung", provides an interesting addition to how resident microbes in the airway help protect against pathogenic microbes. It details how *Prevotella melaninogenica* (Pm) protects against *Streptococcus pneumoniae* (Sp) lung infection. Overall, I found the study to be thorough and well controlled. I think it adds several important findings to the field. I have the following points:

- 1) There are a few additional controls which would support the authors' mechanism. One confounder not addressed here is the indirect effect of Pm inoculation on microbial communities already resident in mice. Could the effect of Pm be indirect through its modulation of abundance of other organisms already resident in the upper airway or lung? This could be address by using antibiotic treatment in their existing models.

We agree that the potential influence of the endogenous microbiome on the protective effect mediated by *P. melaninogenica* is an important question. To address this, we conducted new experiments using live *P. melaninogenica* in both antibiotic-treated and Germ-free mice. In both cases, live *P. melaninogenica* significantly enhanced protection against *S. pneumoniae* lung infection as shown by improved pneumococcal clearance from the lung [Fig. 4a-b]. We also confirmed that *P. melaninogenica*-mediated protection in Germ-free mice was associated with increased neutrophil recruitment and neutrophil expression of TNF-alpha by flow cytometry [Fig. 4c]. These results suggest that *P. melaninogenica* is sufficient to improve protection against *S. pneumoniae* lung infection in the absence of the endogenous microbiome.

2) Could any of the effects be driven from the gut? More specifically, are any of the Pm inoculated via the different routes used in this study getting to the gut and signalling from there?

A recent study published using the same method of i.t. instillation (from the lab in which corresponding author Dr. Clark was trained in this method) demonstrated using Trypan Blue dye that liquid injected i.t. doesn't reach the stomach, as dye was not detected in stomach tissue, unlike mice injected by oral gavage where the dye was clearly visible in the stomach [Bortell *et al* 2021, PMID:33878120]. This suggests that the large majority of the *P. melaninogenica* inoculated directly into the lung (i.t.) is not getting into the gut. While it is unclear whether the live *P. melaninogenica* instilled into the lungs are capable of spreading systemically after inoculation, it is unlikely that the heat-killed preparations of *P. melaninogenica* spread beyond the lung, given the Trypan Blue data. In contrast, *P. melaninogenica* inoculated intranasally may reach the gut, as a portion of this inoculum is likely swallowed by the animal. While intranasal inoculation was not our primary model, we note that the phenotype is similar to i.t. inoculation of heat-killed *P. melaninogenica* [Supplementary Fig. 1b], suggesting that any priming due to bacteria reaching the gut has a limited impact in this setting.

3) Does this effect extend to other Prevotella? The authors only tested a very limited repertoire of other commensals.

We agree that the extent to which this protective mechanism is a conserved feature of airway *Prevotella* species is an interesting and important question. To address this, we conducted new experiments with the following additional live airway *Prevotella* isolates: another strain of *P. melaninogenica* (strain D18), *Prevotella buccae*, *Prevotella tanneriae*, *Prevotella nanceiensis*, and *P. intermedia*. Among these, all except *P. intermedia* were protective against *S. pneumoniae* infection [Fig. 9a]. *P. intermedia* is a major periodontal pathogen, and previously reported to enhance susceptibility to type 4 *S. pneumoniae* [Nagaoka *et al* 2014, PMID: 24478074]. These results indicate that several airway *Prevotella* species are capable of enhancing protection against *S. pneumoniae*, while more pathogenic species such as *P. intermedia* are not protective. To further investigate how these different *Prevotella* species activate neutrophils, we stimulated WT versus *Tlr2*^{-/-} neutrophils with heat-killed preparations of each *Prevotella* isolate. We found that all protective *Prevotella* strains induced neutrophil secretion of TNF-alpha and IL-10 in a TLR2-dependent manner [Fig 9b]. In contrast, *P. intermedia* activated neutrophils in a TLR2-independent manner. These data build on findings presented in the original manuscript highlighting an important role for TLR2-dependent neutrophil activation, suggesting that activation of this innate immune pathway is a conserved feature among protective airway *Prevotella* species.

4) The analysis of the TNF/neutrophil axis is good. I think it's important to understand what the TNF levels in the lung are during neutrophil depletion. This would help to understand whether neutrophils are the sole lung TNF source or not.

To address this question, we depleted neutrophils in *P. melaninogenica*-exposed mice and compared BAL levels of TNF-alpha to those in *Prevotella*-exposed mice treated with isotype control antibody and naïve controls. Neutrophil depletion significantly reduced total TNF-alpha in the BAL of mice exposed to *P. melaninogenica* [Supplementary Fig. 2h]. However, there is still some TNF-alpha in the BAL of neutrophil-depleted mice above that detected in naïve animals, indicating that other cellular sources are responsible for the remaining TNF-alpha induced by *P. melaninogenica*.

5) In the IL-10 KO model, if TNF is neutralized does this rescue the defect caused by IL-10 loss?

The role of TNF-alpha in *Il10*^{-/-} mice was investigated by TNF-alpha neutralization, as suggested. We find that TNF-alpha neutralization had no impact on *S. pneumoniae* burdens 24 h post-infection in *Il10*^{-/-} mice [Supplementary Fig. 7f]. These data indicate either that TNF-alpha neutralization alone was not sufficient to restrain inflammation in these mice, or that too much TNF-alpha abrogation limited the protective immune activation phenotype (which requires TNF-alpha). Regardless, in the absence of IL-10, the modulation of TNF-alpha alone is not sufficient to restore protection. The critical protective versus tissue-damaging responses regulated by IL-10 following exposure to *P. melaninogenica* is an area of ongoing investigation.

6) In the experiments where synthetic TLR2 ligands are administered to mice, they fail to promote Sp clearance promoting the authors to suggest TLR2 is necessary but not sufficient. However, could this be simply a dosing issue? Has a comparable amount of synthetic TLR2 ligand been added to the level of TLR2 agonists produced/displayed by Pm?

To address this question, we first determined that the total protein content of the HK *P. mel.* preparations injected per mouse was 41.9 µg by BCA Protein analysis. Using this value as a reference point for the highest potential amount of TLR2 ligand present, we titrated administration of Pam3SK4, this time using doses of 25-50 µg per mouse. Synthetic TLR2 ligand at the highest dose still failed to induce protection against *S. pneumoniae* [Fig. 5f]. These data support the conclusion that TLR2 agonists alone are not sufficient to mediate protection against *S. pneumoniae*.

7) Are the serine protease inhibitors directly toxic to Sp?

We found that the one-hour exposure to the serine protease inhibitor cocktail used in these studies had no impact on *S. pneumoniae* growth, as measured by CFU/mL [Supplementary Fig. 6c].

8) One of the most striking findings is that it looks like TLR2 is not important for direct recognition of Sp. Care has to be taken with the TLR2^{-/-} mice because direct recognition of Sp could be lost (in addition to recognition of Pm). As a minimum, there must be some statistical comparison in the data shown that confirms, at the point of infection shown in most of the data (24h) TLR2 is not important for Sp defence and clearance.

Analysis of pooled CFUs from three independent experiments comparing WT vs *Tlr2*^{-/-} mice at 24 hpi (n= 9 mice/group) confirms there is no difference in burdens of *S. pneumoniae* [Fig. 6c], as measured by either a Kruskal-Wallis test or individual two-tailed t test. These results are consistent with the findings of other studies which similarly reported no difference between WT and *Tlr2*^{-/-} mice in clearance of WT *S. pneumoniae* from the lungs at 24 hpi (Dessing *et al*,

PMID: 17711480, Lammers et al, PMID: 22721450), and another study that reported no impact on survival (Albiger *et al*, PMID: 17004992). While TLR2 is involved in recognition of *S. pneumoniae*, and *Tlr2*^{-/-} mice have a reduced lung inflammation score during *S. pneumoniae* infection (Knapp *et al*, PMID: 14978119), redundancy with other TLRs which also recognize *S. pneumoniae* and/or successful *S. pneumoniae* evasion of TLR2-induced responses may account for the lack of a clearance defect in *Tlr2*^{-/-} mice at this timepoint.

9) *Tlr2*^{-/-} should be used instead of TLR2^{-/-} for mouse data.

Corrected as recommended.

Reviewer #2 (Remarks to Author):

Review Horn et al - Nat Communications Jan 2022
Airway *Prevotella* promote TLR2-dependent neutrophil activation and rapid clearance of *S. pneumoniae* from the lung
Kadi J. Horn, Melissa A. Schopper, Sarah E. Clark*

This study investigates the potential interaction of *Prevotella melaninogenica*, one of the most commonly isolated species of the lung microbiome, with the bacterial pathogen *Streptococcus pneumoniae*, the most common cause of community-acquired pneumonia.

A clear point is made on associations between the presence of *Prevotella* spp. and the induction of a milder inflammation (sub-clinical with neutrophil involvement) in the context of different pathogenic infections. Microbiome analyses have indicated that increased abundance of *P. melaninogenica* correlates with reduced infections.

Using a mouse co-infection model, the presented data show that *P. melaninogenica* induced an innate immune that resulted in some immune protection against *S. pneumoniae* infection through recognition via TLR2, induction of TNF α , and neutrophil activation. Increased clearance of *S. pneumoniae* from the lung was due to increased serine protease-mediated killing resulting in improved survival.

The results presented here are highlighting the interactions of co-infections in the activation of the immune system. While a suggestion can be made regarding the importance of the whole microbiome in this process, as the study focused on a co-infection model consisting of two bacterial species, this should be made clear in the abstract and the conclusion.

This would take nothing away from the importance of the data but highlights that with our current methods we are not yet able to investigate the whole microbiome interactions in this manner.

We thank the reviewer for these comments. While our revised manuscript expands the number of airway *Prevotella* species associated with increased protection against *S. pneumoniae* in our model, we agree that the focus remains on specific members of the airway microbiome, rather than the microbiome as a whole. As suggested, we clarified in the abstract and conclusion sections that our studies indicate airway *Prevotella* species, specifically, as important members of the respiratory tract microbiome for enhanced protection against *S. pneumoniae* infection in the lung [pg 1, line 25, pg 4, lines 7-10, pg 18, lines 10-13, and pg 19, line 2].

Overall, the presented work is highly relevant to respiratory health and emphasizes the importance of co-infections (the microbiome?). Experiments are thoroughly performed, however, there are too many graphs (some show duplicate results). The manuscript would also benefit from some clearer and shorter description of the results.

We carefully revised the results section to improve the clarity and brevity of our analyses. Graphs showing information similar to that already presented elsewhere have been moved to Supplementary figures. We note that all such graphs contain data from independent experiments, confirming foundational observations made throughout the manuscript. All flow cytometry data are now presented in a more simplified manner, with graphs showing total cell numbers which were next to similar graphs of cell percentages moved to Supplementary figures. We thank the reviewer for this suggestion, as these changes have improved the clarity of the main Figures.

Methods:

Various bacterial strains were used throughout the study. Most of them were grown in their specific medium or broth. Would the medium the strain was propagated in have any effect on its virulence?

Please comment.

We are unaware of any media-dependent impact on virulence for the strains used in this study. We have grown *S. pneumoniae* in several different types of nutrient broth (e.g., Brain Heart Infusion with or without 1% Tween80, Todd Hewitt Broth with Yeast Extract) with similar results. For example, 24 h lung burdens from *S. pneumoniae* grown in Brain Heart Infusion with 1% Tween80 versus Todd Hewitt Broth with Yeast Extract are the same. Some, but not all, of the other bacterial strains used in these studies were grown in one of these media types (e.g., *Corynebacterium* species and *S. salivarius*). *E. coli* and *Prevotella* isolates were only grown using one type of media, which differed from that used for *S. pneumoniae*, however we report similar results between live and inactivated bacteria for both of these strains, suggesting a minimal impact of media on strain virulence. Of note, all bacteria prepared for inoculation into mice were resuspended in PBS, rather than inoculated in the original growth media, which also serves to minimize differences due to growth media.

Animals: Please confirm that there is no potential mispairing of C57BL/6 sub-strains (here B6.129Tlr2tm1Kir (TLR2^{-/-}), and B6.129il10tm1Cgn (IL-10^{-/-} 4) and the C57BL/6J (WT) wild-type strain.

Please comment and add into the manuscript.

Both of the mutant mouse strains used in these studies (*Tlr2^{-/-}* and *Il10^{-/-}*) are on the C57BL/6J genetic background, which is the same as the WT strain [clarified in **Methods, pg 19, lines 18-19**]. The nomenclature for these strains refers to the original gene disruptions, which were generated using strain 129 embryonic stem cells injected into C57BL/6J blastocysts. Heterozygotes were then backcrossed to C57BL/6J for at least 10 generations, maintained at the Jackson Laboratory from which they were purchased for this study.

Prevotella aspiration was modelled in mice by instilling *P. melaninogenica* intratracheally (i.t.) prior to challenge with *S. pneumoniae*. The overall rationale for this was that the murine airway microbiome contains both *Prevotella* and *Streptococcus* species, but *P. melaninogenica* and *S. pneumoniae* are not resident members.

While the model allows for the addition of the required species to the existing lung microbiome, how does such addition change the whole microbiome (not only the response to each other, inflammatory immune responses, or bacterial clearance)? Can the overall effect be attributed to the interaction of *P. melaninogenica* with *S. pneumoniae* or with the overall change in the microbiome?

Please comment and add the information to the manuscript.

From response above to Reviewer #1:

We agree that the potential influence of the endogenous microbiome on the protective effect mediated by *P. melaninogenica* is an important question. To address this, we conducted new experiments using live *P. melaninogenica* in both antibiotic-treated and Germ-free mice. In both cases, live *P. melaninogenica* significantly enhanced protection against *S. pneumoniae* lung infection as shown by improved pneumococcal clearance from the lung [Fig. 4a-b]. We also confirmed that *P. melaninogenica*-mediated protection in Germ-free mice was associated with increased neutrophil recruitment and neutrophil expression of TNF-alpha by flow cytometry [Fig. 4c]. These results suggest that *P. melaninogenica* is sufficient to improve protection against *S. pneumoniae* lung infection in the absence of the endogenous microbiome.

Statistical analysis: Experiments are performed at n=4-5 in each experiment. Although the plotted data in the dot plot look as if they are normally distributed, the authors should consider non-parametric tests and descriptive measures for the small numbers investigated.

All data which passed the Shapiro-Wilk normality test were analyzed using parametric tests (ANOVA, t test), while all data which failed this test of normality were analyzed using non-parametric tests (Mann-Whitney U test, Kruskal-Wallis test). Data from infection experiments were pooled rather than showing representative data for Fig. 3b, Fig. 3e, Fig. 4 and Fig. 6c, for n=9-11 mice/group, though these data were not normally distributed due to the CFU limit of detection (relevant for *P. melaninogenica* treated groups) and were analyzed using non-parametric tests regardless. We acknowledge that the flow cytometry data from representative experiments with n=4-5 mice/group which passed the Shapiro-Wilk normality test could alternatively be analyzed using non-parametric tests and descriptive measures. To address this, we included non-parametric analyses for such data in Fig. 2 and Fig. 6 alongside the parametric tests. Specific tests used and individual p values are specified for all figure panels in the Figure Legends. An updated discussion of the statistical analysis, including normality testing, is included in the Methods [pg 26, lines 1-9].

Results:

Pre-exposure (instillation) of heat-killed *P. melaninogenica* increased survival following a lethal dose of *S. pneumoniae* (fig 1b), however, when equivalent doses of *P. melaninogenica* and of *S. pneumoniae* were used only heat killed bacteria mediated the protection.

“Heat-killed *P. melaninogenica* was protective against both serotype 2 *S. pneumoniae*, which spread systemically by 24 hours, and serotype 3 *S. pneumoniae*, which was restricted to the lung (Fig. 1d-e)”

↳ Early clearance of a sublethal dose of *S. pneumoniae* in mice exposed to *P. melaninogenica* correlated with lower burdens in the lung 10 three days later (Supplementary Fig. 1a)

In subsequent experiments only heat-killed *P. melaninogenica* was used. Importantly, heat killed *E. coli* did not mediate protection (Fig. 1f). However, as both bacteria are Gram negative species and TLR4 agonists, it appears that not all Gram-negative bacteria have a similar effect. Analyses of serum cytokines indicated that the protective effect heat-killed *P. melaninogenica* was mediated by the induction of IL-10 and a reduction of TNF α , differences in cytokines only seen in *P. melaninogenica* exposures.

What is the difference between Fig. 1g (“... less TNF α and more IL-10 at 24 hours post-*S. pneumoniae* infection compared to those exposed to *E. coli* HK or *E. coli* LPS ...”) and Fig. 2b (“Mice exposed to *P. melaninogenica* also had increased systemic TNF α and IL-10 compared to mice treated with PBS or *E. coli* LPS”). Can these graphs be combined?

Data in Fig. 1g [now **Fig. 1e**] are from mice pre-exposed to *P. melaninogenica* and infected with *S. pneumoniae*, while data in Fig. 2b are from mice exposed to *P. mel.* HK or *E. coli* LPS alone (no *S. pneumoniae* infection). Text was added to the legend of Fig. 2 [pg 32, line 5] to bring attention to the point that these data are from mice exposed to *P. mel.* or *E. coli* LPS in the absence of *S. pneumoniae* infection, which is also highlighted in the Results section discussing these data [pg 6, lines 13-14]. We also added headers to all serum and flow cytometry data in **Fig. 1, Fig. 2, Fig. 3, Fig. 6, and Fig. 8** (and associated **Supplementary Figures**) denoting whether the data are from infected or uninfected mice to provide additional clarity.

Further analyses of BAL fluid showed a panel of innate immune cytokines (e.g., increase in IL-1 α , IFN γ , MCP1, IL6, MIP-2) and the recruitment of neutrophils to the lungs after *P. melaninogenica* exposure (Ly6G⁺ cells). In contrast to *E. Coli*, *P. melaninogenica* also showed a stronger activation of the neutrophils (Ly6G⁺ / TNF α ⁺ cells). Monocytes were recruited to the lungs by both strains (Ly6G⁻ / Ly6C⁺ cells) and activated (Ly6G⁻ / Ly6C⁺ / TNF α ⁺ cells). This overall suggests a “selective” effect of *P. melaninogenica* on neutrophils, which potentially mediates the immune protection. When Ly6G⁺ cells or TNF α from Ly6G⁺ cells are deleted, the protective effect on bacterial burden ceases.

Investigations into the host and bacterial requirements for *P. melaninogenica*-induced neutrophil TNF α production showed the importance of TLR2 signalling for this response (inhibition of TLRs). Further, TNF α induction was significantly reduced when heat-killed *P. melaninogenica* was treated with lipase to digest bacterial lipoproteins in the cell membrane, which are bacterial TLR2 ligands.

However, of interest is that additional factors appear to be required to mediate the protective TLR2-lipoprotein signalling effect as neither Pam3SK4 (TLR2 agonist) nor purified *P. melaninogenica* lipoproteins reduced *S. pneumoniae* burdens in the lung (Fig. 4f).

TLR2^{-/-} mice showed that TLR2 is required for *P. melaninogenica*-induced neutrophilic activation and TNF α secretion.

In the co-infection model, neutrophil recruitment was lost in *P. melaninogenica*-exposed TLR2^{-/-} mice infected with *S. pneumoniae*, while the recruitment of inflammatory monocytes was maintained.

This is an important finding that should be added to the figures.

We moved the data showing retention of inflammatory monocyte recruitment in *P. melaninogenica*-exposed Tlr2^{-/-} mice infected with *S. pneumoniae* from a Supplementary Fig. to **Fig. 6e**, as recommended.

The effect of IL10 in the regulation of protective effects of TNF α

Using purified neutrophils, *P. melaninogenica* induced the secretion of IL-10 in a dose-dependent manner, which could regulate TNF α production. Compared to the single infections, co-infection of bone marrow derived neutrophils with *P. melaninogenica* and *S. pneumoniae* shows significantly reduced TNF α release. This is also shown as a systemic response in *Prevotella*-exposed mice.

This is not in Fig. 1g, but in fig 7?

Fig. 1g [now **Fig. 1e**] first shows serum TNF-alpha and IL-10 in WT mice exposed to *P. melaninogenica* and infected with *S. pneumoniae*. In Fig. 7 (now **Fig. 8c**), we add to these data by showing serum TNF-alpha and IL-10 in WT vs. *Il10*^{-/-} mice. We moved the serum IL-10 data for this Figure to **Supplementary Fig. 7c**, as we agree this merely confirms loss of serum IL-10

in *IL10*^{-/-} mice and doesn't add information beyond that shown in Fig. 1. The serum TNF-alpha data was kept in **Fig 8c**, as it shows that there is increased systemic TNF-alpha in *IL10*^{-/-} mice exposed to *P. melaninogenica*.

Additionally, BALF cytokines TNF α , IL-6, IL-1 α , IFN β and IFN γ were also reduced in co-infections. This was observed 48h after co-infections, suggesting that a regulator such as IL-10 must be induced, transcribed etc. before an effect can be seen on protein levels.

IL-10 deficient (*IL-10*^{-/-}) animals

Unexpectedly, *P. melaninogenica*-mediated protection against *S. pneumoniae* was lost in *IL-10*^{-/-} mice, which had similarly high burdens regardless of *P. melaninogenica* exposure (Fig. 7b). In *IL-10*^{-/-} mice, serum TNF α was elevated following *P. melaninogenica* exposure, rather than reduced as in WT mice (Fig. 7c). Further, WT mice exposed to *P. melaninogenica* had significantly elevated systemic IL-10 compared to mice infected with *S. pneumoniae* alone (Fig. 7c).

Additional effect of IL-10 derived from other immune cells.

Detailed comments

Line 5, page 2: "Prevotella are frequently [identified / found] among the top three most abundant bacteria detected in the oral cavity and lungs of healthy adults" – add verb

Corrected as recommended.

Line 26, page 5 ff. "... which like *S. pneumoniae* are gram-positive." Please refer to Gram staining as Gram-positive or Gram-negative, acknowledging Dr Gram, the Danish bacteriologist who developed the technique in 1884. (<https://www.newscientist.com/article/2216418-hans-christian-gram-the-biologist-who-helped-investigate-bacteria/>). Please correct throughout.

Corrected as recommended.

Line 14, page 20: "Ly6G antibody (clone 1A8), or ant-TNF α antibody (clone XT3.11) as indicated (Bio X Cell ..." – add 'i' (anti).

Corrected as recommended.

Line 12, Page 21: "... with analytes were detected on the 14 LSR Fortessa X-20 in the ImmunoMicro Flow Cytometry Shared Resource Laboratory ..." Omit 'were'
Please check for typographical and grammatical errors throughout.

Corrected as recommended. Thank you for bringing these errors to our attention. The revised manuscript has been thoroughly checked for typographical and grammatical errors.

REVIEWERS' COMMENTS

Reviewer #1 (Remarks to the Author):

I thought this was a good study originally and I think it has been improved by the revisions. My concerns have been addressed, I'd be very happy to see it published.

Reviewer #2 (Remarks to the Author):

The authors have made all suggested modifications to the manuscript.

Additional data presented make a significant contribution to enhance the overall results.

Response to Referees

May 23rd, 2022

Manuscript: NCOMMS-21-49627A

Title: **Airway Prevotella promote TLR2-dependent neutrophil activation and rapid clearance of Streptococcus pneumoniae from the lung**

Responses to referees:

Reviewer #1 (Remarks to Author):

I thought this was a good study originally and I think it has been improved by the revisions. My concerns have been addressed, I'd be very happy to see it published.

We thank the reviewer for these positive remarks regarding our revised manuscript.

Reviewer #2 (Remarks to Author):

The authors have made all suggested modifications to the manuscript. Additional data presented make a significant contribution to enhance the overall results.

We thank the reviewer for these positive remarks regarding our revised manuscript.